

**A CONSISTENT GLACIER INVENTORY FOR THE KARAKORAM AND PAMIR REGION**
**DERIVED FROM LANDSAT DATA: DISTRIBUTION OF DEBRIS COVER AND MAPPING**
**CHALLENGES**
**Nico Mölg[1], Tobias Bolch[1], Philipp Rastner[1], Tazio Strozzi[2], Frank Paul[1]**
[1] Institute of Geography, University of Zurich, Switzerland
[2] Gamma Remote Sensing, Worbstr. 225, 3073 Gümligen, Switzerland
Corresponding author:
*Nico Mölg, GIUZ,*
*E-mail : nico.moelg@geo.uzh.ch*



**ABSTRACT.**

The knowledge about the coverage and characteristics of glaciers in High Mountain Asia is still incomplete and heterogeneous. However, several applications such as modelling of past or future glacier development, runoff or glacier volume, rely on the existence and accessibility of complete datasets. In particular, precise outlines of glacier extent are required to spatially constrain glacier-specific calculations such as length, area and volume changes or flow velocities. As a contribution to the Randolph Glacier Inventory (RGI) and the GLIMS glacier database, we have produced a homogeneous inventory of the Pamir and the Karakoram mountain ranges using 28 Landsat TM and ETM+ scenes acquired around the year 2000. We applied a standardized way of automated digital glacier mapping and manual correction using coherence images from ALOS-1 PALSAR-1 as an additional source of information, separated the glacier complexes into individual glaciers using drainage divides derived by watershed analysis from the ASTER GDEM2, and separately delineated all debris-covered areas. Assessment of uncertainties was performed for debris-covered and clean-ice glacier parts using the buffer method and independent multiple digitizing of three glaciers representing key challenges such as shadows and debris cover. Indeed, along with seasonal snow at high elevations, shadow and debris cover represent the largest uncertainties in our final dataset. In total, we mapped more than 27'400 glaciers >0.02 km² covering an area of 35'287 ±1209 km² and an elevation range from 2260 m to 8600 m with regional median glacier elevations varying from 4150 m (Pamir Alai) to almost 5400 m (Karakoram); this being largely due to differences in temperature and precipitation. The coverage of glaciers by debris is on average ~5 %, but glaciers >5 km² have often a much higher area share (>10%), making it an important factor to be considered in subsequent applications.

Location of dataset: http://www.geo.uzh.ch/~nmoelg/glacier_inventory.zip

**KEY WORDS.** High Mountain Asia, Karakoram, Pamir, glacier mapping, glacier inventory, DEM, watershed analysis, debris cover, uncertainties



## 1   INTRODUCTION

Glacier outlines and related inventories provide the baseline for climate change impact assessments (Vaughan et al., 2013), numerous hydrology-related calculations that consider water resources and their changes (e.g. drinking water, irrigation, hydropower production, run-off, sea-level rise) (e.g. Pritchard, 2017; Kraaijenbrink et al., 2017; Bliss et al., 2014), climatic characteristics ( Sakai et al., 2015), or modelling of past and future glacier changes (e.g. Huss and Hock, 2015). All of this also applies to most catchments of High Mountain Asia (HMA), although some of their glacier melt water does not directly contribute to sea-level rise as related rivers end in endorheic basins (e.g. Tarim basin, Aral Sea basin).

By using glacier outlines of partly poor quality, related hydrologic calculations at catchment scale (e.g. Immerzeel et al., 2010) are associated with higher uncertainties. This situation significantly improved during the past years, when several large-scale glacier inventories for HMA had been published, among others the 'Glacier Area Mapping for Discharge from the Asian Mountains' (GAMDAM) inventory (Nuimura et al., 2015) and the new Chinese Glacier Inventory (CGI, Guo et al., 2015). Because the currently available version of GAMDAM did only partially consider ice cover at steep slopes and the CGI only covered Chinese territory, a homogeneous basis for precise calculations covering all relevant catchments at large scale is still missing. As both inventories have been combined for version 5.0 of the Randolph Glacier Inventory (RGI) (Arendt et al., 2015; RGI Consortium, 2017), the related regional-scale calculations using this version (e.g. Brun et al., 2017; Kraaijenbrink et al., 2017; Dehecq et al., 2015; Kääb et al., 2015) still comprise uncertainties which stem from outlines of varying quality.

To overcome this situation, several regional-scale studies digitized glacier outlines themselves (e.g. Rankl and Braun, 2016; Minora et al., 2016) to have a better control on data quality. But these again applied different criteria to delineate glacier extents and are thus not comparable to the existing datasets, making change assessment difficult. On the other hand, the Karakoram and Pamir regions are characterized by a high number of surge-type glaciers (Bhambri et al., 2017; Copland et al., 2011; Kotlyakov et al., 2008) with often strong geometric changes over a short period of time (Paul, 2015; Quincey et al., 2015). A precise inventory is key to determine and maybe better understand such changes. Moreover, the large number of partly heavily debris-covered glaciers in the region (Herreid et al., 2015; Minora et al., 2016) results in interpretation differences and is a large source of uncertainty.

The correct delineation of debris is also important for detecting very subtle past glacier changes (Scherler et al., 2011b) and to correctly model future glacier development (e.g. Kraaijenbrink et al., 2017; Shea et al., 2015), as surface mass balance of ice under a supra-glacial debris layer is different from clean ice (Ragettli et al., 2015 and 2016; Brock et al., 2010; Nicholson and Benn, 2006). The information on debris extent should thus be included in large-scale glacier inventories (Kraaijenbrink et al., 2017).



The main objective of this study is to present a consistent dataset of glacier coverage for the larger and
heavily glacierised mountain ranges Pamir Alai, Western and Eastern Pamir, and the Karakoram of HMA
(Figure 1) along with the spatial distribution of debris cover for the years around 2000. In addition, we
present a structured overview of the difficulties related to glacier mapping in this region as well as an
estimate of the respective uncertainties. Key challenges are the entity assignment as many glaciers in
this region are of surge-type and tributary glaciers can be either connected or disconnected to a larger
main glacier, the already mentioned mapping of debris-covered glacier ice and the differentiation of
debris-covered glaciers from rock glaciers that are increasingly abundant towards the north and the drier
east of the study region.
## 2   Study region
### 2.1   Location and glacier characteristics
The study area comprises a major share of the western part of HMA (Figure 1). It stretches over
~300'000 km² and fully covers the mountain ranges of Pamir and its northern neighbour Pamir Alai in
Uzbekistan, Turkmenistan, Kyrgyzstan, Tajikistan, Afghanistan and China, and the Karakoram in Pakistan,
India, and China.


*Figure 1 : The study region in HMA covers four mountain ranges. Annotations denote locations of figures in the paper and points*
*of orientation. International borders are tentative only as they are disputed in several regions.*

The mountain ranges reach their highest elevations between 5500 m (Pamir Alai) and more than 8600 m
(Karakoram, Hindu Kush), with the K2 being the highest peak with 8611 m. Glaciers are found from
~2300 m up to the highest peaks. With the central Karakoram and the inner Pamir two of the most
heavily glacierised mountain regions worldwide are part of the study region, including some of the
largest glaciers such as Baltoro, Siachen, and Fedchenko with sizes of about 810, 1094 and 573 km$^2$,
respectively. The study region is heavily glacierised and holds some of the world's largest glaciers. High
peaks and deeply incised valleys create an extreme topographic relief that is also reflected in the
geometry of the glaciers, the majority of which are valley glaciers. In the Pamir, also numerous cirques
are present, and hanging glaciers can be found at high elevations in all regions. Larger, flat, high-altitude
accumulation areas are rare and can only be found for some of the largest glaciers. Due to the steep
terrain, most glaciers are partly fed by avalanches from the surrounding steep valley walls (Dobreva et
al., 2017; Iturrizaga, 2011; Hewitt, 2011; Scherler et al., 2011a). This also causes an abundance of glaciers



with partly or completely debris-covered tongues. Whereas debris cover makes glacier mapping difficult,
the strong geometric changes of surging glaciers create additional challenges for glacier inventory
compilation. Rock glaciers are present in all periglacial environments and are abundant also in our study
region.
**2.2   Climate**
The climate of the study region can be subdivided into two major and independent regimes. According
to Bookhagen and Burbank (2010) the Pamir and the major part of the Karakoram are predominantly
influenced by westerly air flows throughout the year (e.g. Singh et al., 1995); towards the south-east, in
the central and eastern Karakoram, the influence of the Indian Summer Monsoon (approx. June-
September) becomes continuously stronger (e.g. Archer and Fowler, 2004). These two regimes define
both the thermal and the moisture conditions that are relevant for glacier distribution. In general, outer
western areas of the mountain ranges (Pamir Alai, western Pamir, Hindu Kush, south-eastern Karakoram)
receive more precipitation; whereas further inland and to the east (Eastern Pamir, central Karakoram)
the climate becomes drier and more continental (Lutz et al., 2014).
Precipitation amounts vary spatially and seasonally. For most of the study region, the main share of
precipitation falls in winter and spring (Archer and Fowler, 2004; Bookhagen and Burbank, 2010). In
regions dominated by westerlies, winter precipitation is mostly advective with cloud altitudes being
lower due to lower temperatures in winter, leading to wetter western mountain margins and dryer
conditions further to the east. This applies to the Pamir Alai, the central Pamir, and the western/central
Karakoram. Convective precipitation plays an important role in dryer regions and occurs predominantly
in spring and summer (Böhner, 2006). In the Eastern Pamir and also the central/eastern Karakoram, the
precipitation peak is shifted towards spring, with important annual precipitation shares even in summer
(Zech et al., 2005; Aizen et al., 2001; Aizen et al., 1997). In monsoon-dominated regions (the border is
roughly at 77°E, Bookhagen and Burbank, 2010), there is only limited winter precipitation coming from
the west, and a mixture of western disturbances and monsoon dominates in summer (Maussion et al.,
2014; Böhner, 2006; Bookhagen and Burbank, 2010; Archer and Fowler, 2004). Little is known about
temperature and precipitation at the elevation of glaciers. Stations in valley floors exhibit a semi-arid
climate in the deeply incised valleys of the Karakoram but also in the continental Eastern Pamir, with
amounts between 70-300 mm yr$^{-1}$ (Seong et al., 2009; Archer and Fowler, 2004). Precipitation amounts
along the edge of mountain ranges and in high altitudes are largely unknown, but can be substantially
higher ("by a factor of ten": Wake, 1989 ; Immerzeel et al., 2015), which is also suggested by snow station
measurements showing snow accumulations of >1000 mm w.e. around 4000 m in the Hunza Basin
(Winiger et al., 2005).



Archer and Fowler (2004) and Fowler and Archer (2006) found a slight cooling in the upper Indus basin
in the second half of the 20[th] century, combined with an increase in winter and summer precipitation.
Glacier changes (see 2.3) suggest a similar trend in the highest reaches of the central Pamir. For
Tajikistan, representative for large parts of the Pamir, Aalto et al. (2017) found an average temperature
increase of 0.1°C per decade during the past 80 years together with a slight precipitation increase at
higher elevations.
**2.3   Glaciers changes**
Glaciers in the Karakoram have gained considerable attention during the last decade. The 'Karakoram
anomaly' that was introduced by Hewitt (2005), based on the observed unusual behaviour of glacier
termini, is now a major research topic, and numerous studies have investigated the recent and longer-
term evolution of climate, changes in glacier extent and volume, as well as glacier dynamics. These
studies suggest that since the 1970's extent and mass of glaciers in the central Karakoram and the Pamir
have on average hardly changed (Bolch et al., 2017; Bajracharya et al., 2015; Bhambri et al., 2013), which
also applies to the beginning of the 21[st] century (Lin et al., 2017; Brun et al., 2017; Gardelle et al., 2013;
Gardner et al., 2013; Kääb et al., 2012), while glaciers in the mountain ranges of Hindu Kush and Hindu
Raj are mostly retreating (Sarıkaya et al., 2013; Haritashya et al., 2009). However, the patterns of climate-
induced glacier change are not to be confounded with the strong geometric changes observed for the
abundant surge-type glaciers in the region that might occur independent of climatic forcing (Bhambri et
al., 2017; Paul, 2015; Quincey et al., 2015; Rankl et al., 2014; Copland et al., 2011). Glaciers in Eastern
Pamir were on average almost in balance like in the Karakoram (Brun et al., 2017; Holzer et al., 2016). In
the Western Pamir, glacier volume evolution seems to be more negative, but is for the first decade of
the 2000s still relatively modest (Brun et al., 2017); moreover, satellite images of the past two decades
reveal that many glaciers in this region have surged (e.g. Wendt et al., 2017). However, many of the non-
surge-type glaciers in the Pamir are continuously retreating and losing area/mass since the Little Ice Age
(Khromova et al., 2006; Shangguan et al., 2006).
**3   Input Data**
As a mapping basis we have used six Landsat 5 TM and 22 Landsat 7 ETM+ Level 1T scenes, the latter
offering a 15 m panchromatic band for improved mapping quality (Table 1). Additionally, we have also
used coherence images derived from ALOS-1 PALSAR-1 scenes acquired around 2007 to aid in mapping
the debris-covered glacier parts, and the global digital elevation model GDEM Version 2 from ASTER
(GDEM2). The TM and ETM+ scenes served as a basis for glacier mapping while the coherence images
were used for corrections of debris-covered glacier areas. Moreover, satellite images available in Google



Earth served as a visual control for outline detection, with data originating mainly from very high-
resolution optical sensors such as Quickbird, Worldview, Pléiades 1A and 1B as well as SPOT6 and SPOT7
(GoogleEarth 2017); unfortunately these were not available for all regions.
*Table 1 : List of Landsat scenes used to compile the inventory.*

| WRS2 path-row | Date | Scene ID | Sensor | HMA Region |
|---|---|---|---|---|
| 146-036 | 2000-10-08 | LE71460362000282SGS00 | ETM+ | Karakoram |
| 147-035 | 2002-08-02 | LE71470352002214SGS00 | ETM+ | Karakoram |
| 147-036 | 2002-08-02 | LE71470362002214SGS00 | ETM+ | Karakoram |
| 147-036 | 2000-08-28 | LE71470362000241SGS00 | ETM+ | Karakoram |
| 148-035 | 2000-09-04 | LE71480352000248SGS00 | ETM+ | Karakoram |
| 148-035 | 2001-07-21 | LE71480352001202SGS00 | ETM+ | Karakoram |
| 148-036 | 2000-09-04 | LE71480362000248SGS00 | ETM+ | Karakoram |
| 149-033 | 2009-07-26 | LT51490332009207KHC00 | TM | Eastern Pamir |
| 149-034 | 1998-08-13 | LT51490341998225XXX01 | TM | Eastern Pamir, Karakoram |
| 149-034 | 2000-09-11 | LE71490342000255SGS00 | ETM+ | Eastern Pamir |
| 149-035 | 1998-08-13 | LT51490351998225XXX01 | TM | Karakoram |
| 149-035 | 2001-08-29 | LE71490352001241SGS00 | ETM+ | Karakoram |
| 150-033 | 2000-09-02 | LE71500332000246SGS01 | ETM+ | Eastern & Western Pamir |
| 150-034 | 1998-08-20 | LT51500341998232BIK00 | TM | Eastern & Western Pamir, Karakoram |
| 150-034 | 2000-09-02 | LE71500342000246SGS01 | ETM+ | Western Pamir |
| 150-035 | 1999-09-16 | LE71500351999259SGS00 | ETM+ | Karakoram |
| 151-032 | 1999-09-23 | LE71510321999266EDC00 | ETM+ | Pamir Alai |
| 151-033 | 2000-08-24 | LE71510332000237SGS00 | ETM+ | Western Pamir |
| 151-034 | 2002-08-30 | LE71510342002242SGS00 | ETM+ | Western Pamir |
| 151-034 | 2001-07-26 | LE71510342001207SGS00 | ETM+ | Western Pamir |
| 151-035 | 2002-08-30 | LE71510352002242SGS00 | ETM+ | Karakoram |
| 152-032 | 2000-09-16 | LE71520322000260SGS00 | ETM+ | Pamir Alai |
| 152-033 | 2000-09-16 | LE71520332000260SGS00 | ETM+ | Pamir Alai, Western Pamir |
| 152-034 | 2001-08-02 | LE71520342001214SGS00 | ETM+ | Western Pamir |
| 152-034 | 2000-08-31 | LE71520342000244SGS00 | ETM+ | Western Pamir |
| 153-032 | 2000-09-15 | LT51530322000259XXX02 | TM | Pamir Alai |
| 153-033 | 2000-09-15 | LT51530332000259XXX02 | TM | Pamir Alai, Western Pamir |
| 154-033 | 2000-08-29 | LE71540332000242EDC00 | ETM+ | Pamir Alai |


Coherence images have been produced from ALOS-1 PALSAR-1 scenes usually separated by 46 days and
acquired over summer (Table 2). The processing of the images takes into account a number of effects
(e.g. sensor geometry, radiometric calibration, frequency interference etc.) that influence the noise of
the radar interferogram. The remaining decorrelation can be ascribed to changes of the landscape
properties, i.e. the movement of landforms. More details on the processing line can be found in Frey et
al. (2012).



*Table 2 : List of ALOS-1 PALSAR-1 scenes used to generate the coherence images.*

| Path | Frames | Date 1 | Date 2 | Interval (days) | HMA Region |
|------|--------|--------|--------|-----------------|------------|
| 533 | 770-780 | 20090722 | 20090906 | 46 | Pamir Alai |
| 528 | 720-730 | 20090722 | 20090906 | 46 | Karakoram, Western Pamir |
| 528 | 750-760 | 20090722 | 20090906 | 46 | Western Pamir |
| 522 | 700 | 20070822 | 20071007 | 46 | Karakoram |
| 523 | 690 | 20070724 | 20070908 | 46 | Karakoram |
| 524 | 690 | 20070810 | 20070925 | 46 | Karakoram |
| 524 | 700-710 | 20070810 | 20070925 | 46 | Karakoram |
| 524 | 750-760 | 20070810 | 20070925 | 46 | Eastern Pamir |
| 523 | 690-700 | 20070608 | 20070724 | 46 | Karakoram |
| 525 | 700-730 | 20070712 | 20070827 | 46 | Karakoram |
| 525 | 750-770 | 20070712 | 20070827 | 46 | Eastern Pamir |
| 526 | 710-730 | 20070613 | 20070729 | 46 | Karakoram |
| 526 | 770 | 20070613 | 20070729 | 46 | Eastern Pamir |
| 527 | 710-730 | 20070815 | 20070930 | 46 | Karakoram, Western Pamir |
| 529 | 720-750 | 20070618 | 20070918 | 92 | Karakoram, Western Pamir |
| 529 | 760-780 | 20070618 | 20070918 | 92 | Western Pamir, Pamir Alai |
| 530 | 720-750 | 20070705 | 20070820 | 46 | Karakoram, Western Pamir |
| 530 | 760-770 | 20070705 | 20070820 | 46 | Western Pamir, Pamir Alai |
| 530 | 780 | 20070705 | 20070820 | 46 | Western Pamir, Pamir Alai |
| 531 | 730 | 20070722 | 20071022 | 92 | Western Pamir |
| 531 | 750-770 | 20070722 | 20071022 | 92 | Western Pamir |
| 531 | 780 | 20070722 | 20071022 | 92 | Western Pamir, Pamir Alai |
| 532 | 720-750 | 20070808 | 20070923 | 46 | Western Pamir |
| 532 | 760-770 | 20070808 | 20070923 | 46 | Western Pamir, Pamir Alai |
| 532 | 780 | 20070808 | 20070923 | 46 | Western Pamir, Pamir Alai |
| 535 | 770-780 | 20070705 | 20070820 | 46 | Pamir Alai |
| 536 | 770 | 20070722 | 20071022 | 92 | Pamir Alai |
| 537 | 770 | 20070808 | 20070923 | 46 | Pamir Alai |


A DEM is needed to retrieve drainage divides and topographic information for a glacier inventory. The
freely available SRTM DEM and the GDEM2 (both with 30 m cell size) could have been used for this
purpose. The optical GDEM2 has potentially a reduced quality in low contrast regions such as shadow
and snow-covered accumulation regions, but it has been averaged from scenes acquired over a 12-year
period strongly reducing these factors. On the other hand, the SRTM DEM has a precise acquisition date
(February 2000), but suffers from data voids in steep terrain due to radar shadow and layover, which
affect the final quality over glacierised areas, in particular when using void-filled versions. A direct
comparison (subtraction) of both DEMs as recommended by Frey and Paul (2012) confirmed these
differences. We finally decided working only with the GDEM2 as it had much less data voids along
mountain crests (important to derive correct drainage divides) and because it is spatially consistent, i.e.
data voids over glaciers in the SRTM DEM did not have to be filled with some other DEM data (which is



beneficial for deriving consistent topographic information and increases traceability). The vertical
accuracy was found to be around 9 m (probably higher in steep terrain) and similar for both DEMs (Satgé
et al., 2015). For consistency, the glacier separation as well as all subsequent topographic analysis of
glacier elevation, slope and aspect are thus based on the GDEM2.

## 4 Methods

### 4.1 Glacier mapping

We applied the well-established semi-automatic band ratio method (Paul et al., 2002) to classify glaciers
(the clean-ice and snow part), taking advantage of the reflection contrast between snow/ice and other
land surfaces in the red and shortwave infrared (SWIR) parts of the electromagnetic spectrum,
corresponding to Landsat TM or ETM+ bands 3 (red) and 5 (SWIR). An individual, scene-adjusted band
ratio threshold between 1.5 and 3.5 is applied to separate glaciers and snow from other terrain and
compute a binary raster image, which has been smoothed using a 3 by 3 majority filter and is then
converted to a vector file for further editing.
Due to the spectral similarity of debris on and off glaciers, there is so far no method available to
automatically map debris cover over a large set of glaciers using optical satellite imagery alone. Hence,
several studies have tested combined approaches that generally include topographic information
derived from a DEM and other data (Robson et al., 2016; Racoviteanu and Williams, 2012 ; Rastner et
al., 2014 ; Bolch et al., 2007 ; Paul et al., 2004). However, all methods require time-consuming manual
post-processing, and the quality of the results depends to some extent on the experience of the analyst.
As debris-covered glacier tongues can be difficult to identify visually, even when using high-resolution
images (Paul et al., 2013), we have utilized coherence images to aid in the interpretation of stable versus
non-stable terrain. Such images have also been used for glacier mapping in Alaska (Atwood et al., 2014)
or as supportive means for correcting automatically derived glacier outlines in the western Himalaya by
Frey et al. (2012).
The elevation of a glacier can be described by different elevation parameters. One that is well suited for
a comparison between different glacier types and sizes as well as an indication for climatic differences is
the median elevation which is indicative of the ELA at a balanced mass budget (Braithwaite and Raper,
2009) and similar to the mid-point elevation (Raper and Braithwaite, 2009), that has been used in several
studies to characterize glaciers (e.g. Haeberli and Hoelzle, 1995) and climatic conditions, primarily
precipitation amounts (e.g. Sakai et al., 2015; Bolch et al., 2013).



### 4.2    Mapping challenges and solutions

The main challenges for mapping glaciers in this region are the correct delineation of debris-covered glacier parts (including their separation from rock glaciers), seasonal snow, cast shadow, and orographic clouds. In the following, we shortly describe these challenges and present the techniques applied to overcome them.

**Debris cover:** The main reason for extensive debris cover on glaciers is steep/high topography with ice-free rock walls leading to rock falls and avalanches onto the glacier surface (e.g. Herreid et al., 2015; Scherler et al., 2011a; Paul et al., 2004). Apart from the central Karakoram, most regions exhibit glacier recession, which is another factor for increasing debris coverage on the glacier surface (Rowan et al., 2015; Kirkbride and Deline, 2013).

The debris-covered glacier area in this study was mapped manually by editing the automatically derived clean-ice outlines. Key difficulties in identifying these regions are: the small solar incidence angle at these latitudes (reducing topographic contrast in the terminus region), unclear boundaries between supra-glacial debris and moraines or rock glaciers, and debris in shadow (e.g. Bishop et al., 2014). Heavily debris-covered glacier tongues are often in contact with lateral or frontal moraines (Figures Figure 2, Figure 3 and Figure 4) and their composition is very similar leading to similar spectral properties and the need of applying other measures for identification. Whereas human recognition has the ability to trace very subtle features for identification of debris on glaciers, the 30 m spatial resolution of Landsat images is often too coarse for a clear assignment.

In this study we mostly relied on the ALOS-1 PALSAR-1 coherence images for identifying the margins of debris-covered glaciers (Figure 2). Their usability decreases with decreasing glacier size and when the images become "fuzzy" and glacier margins less clear. Additionally, the terrain surrounding glaciers is not always stable enough to distinguish between glaciers (=moving) and glacier-free terrain (=stable). In particular, permafrost landforms such as rock glaciers, steep scree slopes, and moraines are also moving, making it difficult to define the boundary between glaciers and rock glaciers or ice-cored moraines (Figures Figure 2, Figure 3 and Figure 4). In such cases we also used the very high-resolution imagery available in Google Earth and similar tools for identification. Furthermore, multi-temporal data aided in terminus identification by either providing better contrast or by using them in animations (Paul, 2015). Finally, we also applied glaciological background knowledge to check whether small glaciers with unusually long tongues that have a small accumulation area, or the importance of snow avalanching to the accumulation area of a glacier are considered to determine the expected size.




*Figure 2 : Mapping of heavily debris-covered tongues using PALSAR-1 coherence images.*


*Figure 3 : Elongated rock glaciers that are (almost) connected to the active glacier tongue are hard to distinguish. PALSAR-1*
*coherence images are not decisive in this case, but high resolution imagery is.*


*Figure 4 : Extensive moraines and large areas of debris can be found on dead ice and active glaciers.*

In contrast to glaciers – massive bodies of ice originating from continuous snow accumulation – rock
glaciers have a different genesis: they develop in a permafrost environment either from ice-cored
moraines or on talus slopes that provide constant debris input, and commonly have a higher debris
content than glaciers (Berthling, 2011; Haeberli et al., 2006; Barsch, 1996). Especially towards the
cold/dry regions of central Asia, rock glaciers of both types are increasingly abundant (Bolch and
Gorbunov, 2014; Gorbunov and Titkov, 1989). In particular moraine-derived rock glaciers challenge the
analyst as there is often a continuous transition between the glacier and the rock glacier, making it hard
to define a divide (cf. Monnier and Kinnard, 2015). A well-developed rock glacier can in principle be
distinguished from a debris-covered glacier by characteristic surface patterns such as the arc-shaped
transverse ridge and furrow structure instead of the longitudinal debris striations and supra-glacial
ponds found on most debris-covered glaciers (Bishop et al., 2014; Bodin et al., 2010). However,
identifying such differences using remotely sensed imagery requires a spatial resolution better than 15 m
and might not work at all when rock glaciers are not well developed. We separated debris-covered
glaciers from rock glaciers based on interpreting the above data sources (Google Earth, coherence
images) and their known morphological characteristics. In the case no clear boundary could be found,
we followed a more conservative interpretation that might have resulted in a potential underestimation
of the debris-covered glacier area.
**Seasonal snow:** Seasonal snow can obscure the underlying glacier ice and is included in the automatic
classification result due to the similar reflection properties of snow and ice. Seasonal snow and clouds
also required consideration of scenes from other years than 2000. For larger glaciers with a low-lying
terminus, it would have been possible to adjust the (snow-free) terminus to the year 2000 scene; we
have not applied this in favour of temporally consistent glacier outlines. Interestingly, for some regions
it was much harder to find satellite scenes with satisfying snow conditions than for others. It was



particularly difficult for the Eastern Pamir and some parts of the northern/central Karakoram, potentially
resulting in related higher area uncertainties for the accumulation areas of the glaciers. Our strategy to
reduce the impact of wrongly mapped seasonal snow was threefold: we applied a size filter of 0.02 km²
to remove the smallest snow patches; snow attached to glaciers was manually removed after visual
inspection, and in some regions a different scene (with better snow but maybe less good cloud
conditions) was chosen to improve results (see Table 1). Despite these measures, we assume that glacier
area in this inventory is likely overestimated due to the inclusion of seasonal snow.
**Shadow:** Cast shadows from mountains decrease reflection values, partly down to near-zero. This results
in considerable noise in this region for a band ratio using a red (or near infrared) band. As TM band 1
(blue) is strongly influenced by atmospheric scattering, ice and snow in shadow are much better visible
and can be distinguished using an additional threshold (e.g. Paul and Kääb, 2005). Although the shadow
problem is less pronounced in lower latitudes due to the higher solar elevation angle, it is still a problem
in the study region due to the high and steep terrain. We have used the additional blue band to map
glaciers in shadow automatically or applied manual corrections on contrast enhanced true-colour
composites in case the automated refinement was not successful. We also analysed scenes from a
different date or another sensor (incl. very high resolution imagery as available in Google Earth and
similar tools) to reveal if glaciers are possibly present in this region (Figure 5). However, this is time-
consuming and in some regions images are not available or do not meet the criteria for glacier
identification (e.g. due to snow cover). In the case glaciers in shadow could not be identified, a related
underestimation of glacier area results.


*Figure 5 : Glacier detection in shadow with the supporting input of high resolution Google Earth images.*

**Cloud coverage:** Cloud-free scenes have been available for most regions. In the few cases when cloud
cover prevented glacier mapping, the problem was solved 'multi-temporally' by using additional scenes
from years close to the year 2000 (Paul et al., 2017). In some regions, scenes with high cloud coverage
and possible precipitation events were followed by scenes with extensive snow coverage, so that we had
to use scenes from other years. The entire study region is thus a mosaic of many individual scenes (see
Table 1).
### 4.3    Calculating the debris-covered area share of glaciers
For calculating the area share of debris cover, we decided to consider only the ablation areas of glaciers
(i.e. the region below their median elevation), because debris deposited in the accumulation area should



emerge on the glacier surface only below the ELA (Braithwaite and Raper, 2009; Braithwaite and Müller,
1980). We distinguished the debris cover from snow and ice surfaces by applying a constant threshold
of 2.0 to all band ratio images from Landsat TM bands 3 and 5 (red and SWIR) and subtracted the
resulting clean-ice glacier map from the corrected glacier map. The threshold was found empirically with
satisfying results for all scenes from TM and ETM+ sensors. Changing the threshold by ±0.2 changed the
result less than the mapping uncertainties (~5% for debris-covered areas, see chapter 6).


*Figure 6 : Debris cover classification in the Kongur Shan in the Eastern Pamir.*

**4.4     Glacier definition and separation using drainage divides**
We based the mapping and division of glaciers on the GLIMS definition of a glacier (Raup and Khalsa,
2007), stating that a glacier includes 'all tributaries and connected feeders that contribute ice to the
main glacier, plus all debris-covered parts of it'. Stagnant ice masses (e.g. from a former surge) that were
still connected to the glacier tongue were mapped as part of the glacier. In case the active glacier has
clearly receded away from the stagnant 'dead' ice (e.g. after a surge phase), only the active glacier was
mapped. In contrast to this definition, the surging Bivachny glacier tongue was separated from
Fedchenko glacier in the confluence region although one can argue that Bivachny is a connected feeder
(see Fig. 6 in Wendt et al., 2017). A size filter of 0.02 km² was applied to remove seasonal snow fields
and remaining noise. Snow fields, be it seasonal or perennial and glaciers larger than this are considered
as glaciers.
Glacier complexes – at least two glaciers connected in their accumulation areas – can be split into single
glaciers using drainage divides derived from a DEM. This is performed in two-steps. Firstly, raw drainage
basins are calculated by watershed analysis using a flow-direction grid derived from a sink-filled DEM.
Afterwards, overlying raw basins are merged to one basin polygon per glacier considering pour points
and a buffer (Falaschi et al., 2017; Kienholz et al., 2013; Bolch et al., 2010). This approach proved to be
robust even for the large regions of the Karakoram and Pamir as in general very steep mountain crests
divides glaciers. Secondly, manual corrections were performed which took about 90% of the total
processing time. Gross errors were improved using a colour-coded flow direction grid in the background,
a hillshade, the original Landsat scenes and sometimes oblique views in Google Earth. We assigned
separate parts of a glacier to the same glacier ID if these parts were obviously linked by mass transport
('regenerated glaciers').



### 4.5     Uncertainty estimation

Since there is no "ground truth" or reference data for any larger set of glaciers in the study region, we calculated uncertainties for the relevant input data rather than accuracy (Paul et al., 2017).

Glacier mapping uncertainties originate from the coverage of glaciers by seasonal snow and/or debris, shadow and clouds. These need to be corrected manually (on-screen digitizing) by a well-trained analyst. According to the literature, the uncertainty of automatically and manually digitized glacier outlines (clean ice only) ranges between 2 and 5% and is dependent on glacier size (Paul et al., 2013; Paul et al., 2011 ; Andreassen et al., 2008 ; Bolch and Kamp, 2006). Paul et al. (2013) estimated uncertainties using a sample of manually and automatically digitized glaciers from a number of experts and found a mean standard deviation of ~5%. Other studies (Bolch et al., 2010; Granshaw and G. Fountain, 2006) have used a buffer-based estimate, where the final uncertainty depends on the pixel size of the input image. The study by Paul et al. (2017) suggested a tiered system of uncertainty assessment related to workload. We used three of the methods: (1) fixed uncertainty values applied to all glaciers, (2) the buffer method with different buffer sizes for clean and debris-covered glacier parts, and (3) independent multiple digitization of outlines by all analysts for three difficult debris-covered glaciers.

For (1), we applied an uncertainty of ±2% for the clean ice and ±5% for the debris covered ice. This is an upper boundary estimate, because it does not account for the overlapping area of the two surface types. For the buffer method (2) we applied an uncertainty of ±½ pixel for clean-ice parts and ±1 pixel for debris-covered parts. This also provides an upper-bound estimate and we use the standard deviation of the uncertainty distribution for the estimate, as a normal distribution can be assumed for this type of mapping error. It is applied to glacier complexes excluding overlapping areas as well as the border of clean and debris-covered ice of the same glacier. Due to the abundant debris-covered glaciers in the study region, we also performed method (3) to obtain a more realistic uncertainty estimate for the analysts participating in the outline correction. They manually corrected three times the outlines of three example glaciers from different regions with differing additional information being considered (e.g. coherence images and Google Earth imagery). The glaciers are of different size and contain a substantial debris-covered part, combined with difficulties of moraines, glacier confluences regions, and cast shadow.

As not all satellite scenes used to compile the inventory are from the same year, there is a certain temporal uncertainty introduced. However, glacier changes within the ±2 year difference to the target year 2000 are likely within the uncertainty of the glacier outlines and should thus not matter. The actual date information is given for each glacier in the attribute table.



## 5 Results

### 5.1 Basic statistics



We identified 27'437 glaciers (larger 0.02 km²) in the four HMA regions covering 35'287 ±1209 km².
Western Pamir and Karakoram host each over 10'000 glaciers whereas the other regions contain 2000-
4000. As in other larger regions where detailed glacier inventories have been compiled (e.g. Kienholz et
al., 2015; Guo et al., 2015; Pfeffer et al., 2014; Le Bris et al., 2011; Bolch et al., 2010) , the histogram is
strongly skewed towards small glaciers (see Figure 7).


*Figure 7 : Histogram of all glaciers by number. Please note the logarithmic scale of the y-axis.*

Only 3.5% (985) of all glaciers are larger 5 km², most of them are located in the Karakoram. In total, they
cover over 60% of the glaciericed area. On the other hand, 83% (23048) of all glaciers are smaller than 1
km² but cover only ~15% of the total area. The mean glacier size is 1.29 km², with large differences
between the regions: from 0.57 km² in the Pamir Alai to 2.07 km² in the Karakoram (Table 3). The average
median elevation is 4978 m and 5169 m for all glaciers and glaciers larger 5 km², respectively, and differs
only few metres from the mean elevation.

*Table 3 : The upper table shows the basic inventory statistics for all glaciers, the lower table only for glaciers larger 5 km².*

| All glaciers | | | Mean of | | | | | |
|---|---|---|---|---|---|---|---|---|
| | Glacier area (km²) | Uncertain-ty (km²) | Glacier area (km²) | Max. elev. (m) | Median elev. (m) | Min elev. (m) | Slope (°) | No. glaciers |
| **All** | 35519.7 | ±1209 | 1.29 | 5238 | 4978 | 4723 | 26.4 | 27877 |
| **Pamir Alai** | 2072.5 | ±71 | 0.57 | 4359 | 4147 | 3962 | 25.1 | 3655 |
| **Pamir West** | 9464.4 | ±324.3 | 0.85 | 5105 | 4871 | 4654 | 25.5 | 11098 |
| **Pamir East** | 2278.9 | ±78.1 | 0.98 | 5305 | 5049 | 4801 | 26.4 | 2326 |
| **Karakoram** | 21694.6 | ±735.6 | 2.01 | 5693 | 5392 | 5076 | 27.7 | 10798 |

| Glaciers ≥5km² | | | Mean of | | | | | |
|---|---|---|---|---|---|---|---|---|
| | Glacier area (km²) | Uncertain-ty (km²) | Glacier area (km²) | Max. elev. (m) | Median elev. (m) | Min elev. (m) | Slope (°) | No. glaciers |
| **All** | 22269.0 | ±763.0 | 22.6 | 6134 | 5169 | 4244 | 22.6 | 985 |
| **Pamir Alai** | 671.6 | ±23.0 | 12.9 | 5024 | 4111 | 3379 | 19.5 | 52 |
| **Pamir West** | 4752.0 | ±162.8 | 17.3 | 5806 | 4882 | 4107 | 21.7 | 275 |
| **Pamir East** | 1090.5 | ±37.4 | 15.1 | 6342 | 5196 | 4204 | 25 | 72 |
| **Karakoram** | 15754.9 | ±539.8 | 26.9 | 6361 | 5394 | 4389 | 23 | 586 |




### 5.2   Extremes

In the Pamir Alai, the largest glacier is Zeravshan glacier with an area of 106.3 ±6.7 km², three times larger than the second largest. Zeravshan glacier stretches over 2600 m from 2800 m to 5400 m, close to the highest elevations in the Pamir Alai range. The largest elevation range is covered by Tandykul glacier (39°27′N, 71°8′E) 50 km further east, with almost 3000 m (2450-5400 m). Its heavily debris-covered tongue lies in a deep valley that is well shielded to the south. Overall, only a few larger valley glaciers (19 larger 10 km²) have several large tributaries.

In the Western Pamir, Fedchenko glacier is by far the largest with 573 ±19.5 km² (not including Bivachny Glacier at 170 ±8.5 km²). Bivachny glacier starts right below the summit of Pik Ismoi Somoni (formerly known as Pik Communism, 7495 m) and terminates at about 3420 m, whereas Fedchenko glacier stretches from Pik Abuali Ibn Sino, 6940 m down to below 2900 m, hence both glaciers are spanning an elevation range of over 4000 m. The region hosts several large glacier systems (13 larger 50 km², 108 larger 10 km²) that are arranged in two clusters: One is around Fedchenko glacier in the Yazgulem Range and one around Pik Lenin in the Trans-Alai Range. Also this region has steep topography and several glaciers reach an elevation range of around 4000 m. However, these numbers are a snapshot in time and have to be treated with care, since there are many surge-type glaciers whose current phase state can significantly influence minimum elevation and area (Kotlyakov et al., 2008). We found the lowest-lying terminus at a very small-sized, north-facing and likely avalanche-fed glacier (70.65°E/38.99°N) in the Petra Pervogo Range, reaching down to below 2400 m.

The Eastern Pamir region has 38 glaciers larger 10 km², evenly distributed over the individual mountain ranges. The largest glacier (109.4 ±6.9 km²) is Karayaylak glacier draining the northern basin of the Kongur Shan. It starts at the top of Kongur Tagh, with 7680 m the highest mountain in the Pamir, and reaches down to 2819 m, spanning an elevation range of over 4800 m, which is by far the largest value in this region. One of its tributaries has reportedly surged in 2015 (Shangguan et al., 2016). Neighbouring Qimgan glacier starts at the same peak facing south-east and reaches down to 3160 m (almost 4500 m elevation range). A smaller, east-facing glacier in the Oytagh glacier park reaches the same low elevation as Karayaylak glacier (2824 m).

Siachen glacier is the largest of its kind in the Karakoram. With an area of 1094.2 ±31.2 km² (including all of its major tributaries) it is by far the largest glacier in the study area and with over 70 km length Siachen and Fedchenko are the longest glaciers in the mid-latitudes. Two more glaciers have an area over 500 km²: Baltoro: 810 ±36.1 km² and Biafo: 560 ±23.8 km². Both glaciers reach their lowest elevations in the central part of the Hunza valley (around Gilgit), with terminus elevations of around 2500 m and below (Hopar glacier: 2260 m). Two large glaciers reach elevation ranges of 5200 m (Batura and Baltoro),




but also smaller glaciers like Shishper (45 ±4.1 km²), Pasu (62.2 ±1.7 km² and Rakaposhi (14.4 ±0.9 km²)
stretch over an elevation range of 5000 m. Once again, many of these glaciers are of surge-type (e.g.
Bhambri et al., 2017), and their minimum elevations and area values after a surge might strongly differ
from those at the end of a quiescent phase. The highest glacierised regions in the Karakoram are found
around K2 (8611 m; Baltoro glacier) and Distichal Sir (7885 m; Yazghil glacier, Hispar glacier).
**5.3  Glacier aspect analysis**
On average, most glaciers are oriented towards the North sector (mean: 71.5% ±5.4%, Figure 8a). The
relative distribution is similar among the regions, and the largest variations occur in the aspects with
small glacier share (SE, S, SW; normalized STDEV: 0.21, 0.32, 0.31). The distribution of single cells (instead
of one mean aspect value per glacier) shows a similar pattern although with less significance of North
aspect. Nevertheless, the North sector has the highest share in all regions (mean: 56.5% ±5.7%, Figure
8b), while South and South-West host the smallest share of glacierised area.
In contrast to other regions, we found no correlation between median elevation and aspect.


*Figure 8 : Glacier orientation of the different HMA regions. (a) shows the values based on average glacier aspect, (b) is based on*
*the 30 m raster cells. Lower elevations tend to have a higher share of north-facing glacier area. The respective numbers of "All"*
*are given in the table (c).*

**5.4  Glacier slope analysis**
The mean slope of all glaciers is 26.4°. It decreases to 22.6° for glaciers larger 5 km², hence mean slope
is size-dependent. The decrease in mean slope between the sample of all glaciers and glaciers larger
5 km² is relatively large for Pamir Alai (-5.6°) and very small for the Eastern Pamir (-1.4°). Mean slope
varies between different parts of the glacier, with the accumulation area being the steepest section and
the debris-covered areas being by far flatter in all regions (Figure 9).


*Figure 9 : Slope per glacier regions and surface type (avg. slp = average slope of glacierised area, avg. slp acc = average slope in*
*the accumulation area, avg. slp abl = average slope in the ablation area, avg. slp deb = average slope in the debris-covered area).*

To determine whether glaciers constantly get steeper from the terminus to the upper reaches of the
accumulation area, we normalized the elevation distribution of all glaciers such that each glacier covered
the value range from 0 to 1 from the tongue to the upper end, divided into sections of 0.1. The result



clearly reveals a mean slope of about 12° in the lower parts and a constant increase to over 30° at the
highest elevations of each glacier (Figure 10). The uppermost band is again somewhat flatter, possibly
due to transition of slope direction at crests. The pattern is similar in all regions, but the slope increase
along the glacier is higher than average in the Eastern Pamir, and lower than average in the Pamir Alai.


*Figure 10 : Glacier slope along ten glacier elevation sections. The glaciers were normalised for elevation to compare high and*
*low elevation glaciers.*

**5.5  Glacier elevation analysis**
The median elevation of glaciers larger 2 km² ranges from 2800 m to over 6500 m. There is a statistically
significant correlation (p<0.001) between median elevation and latitude (R²=0.48) and longitude
(R²=0.66), which appears as a rise of median elevation from North-West towards South-East across the
study region (Figure 11).


*Figure 11 : Glacier median elevation over the study area of glaciers larger than 0.5 km². The inset shows median elevation,*
*standard deviations and minimum and maximum elevations per bin.*

This rise becomes even clearer when looking at separated areas along a 'fishbone' transverse profile of
our study region (inset). The average values of each segment reveal a rise in median elevation from
3980 m (bin 1) to 5860 m (bin 6), with an average trend of 1.9 m km⁻¹ along the profile.
**5.6  Hypsography**
Plotting the glacier hypsography of the different HMA regions (Figure 12a) reveals a number of further
differences among the regions. Most apparent is the difference in elevation: the median elevation
extends from 4141 m (Pamir Alai) to 5419 m (Karakoram), with Western Pamir (4941 m) and Eastern
Pamir (5119 m) in between the two. Most of the glacierised area is located in the Karakoram (60%) where
the ice is distributed over a large elevation range (Figure 12b). In contrast, in the Pamir Alai most of the
glacier area is situated closely around the median elevation. The large glaciers in the Karakoram reach
far down and occupy large areas in lower elevations, further away from the median elevation than in
other regions. Eastern Pamir shows a similar drop in area share of higher elevations, but the curve
flattens in elevations over 1000 m above the median elevation. This is related to the shape of topography



that is dominated by distinct mountain ranges with large areas above 6500 m (Kongur, Muztag Ata,
Kingata Shan). When analysing the hypsography of glaciers with over 10% debris-covered area compared
to the rest of the sample, the insulation effect becomes visible, with debris-covered glaciers occupying
considerably more area at lower elevations (Figure 12c).


*Figure 12 : Glacier hypsography of the different regions (a), normalized by the respective median elevation (b). Dashed lines*
*represent the 25% and 75% area elevations. (c): Hypsography comparison of more and less debris-covered glaciers.*

**5.7   Debris cover**
The mapping quality of the debris-covered areas is defined by the corrected outlines as well as by the
clean-ice threshold used to differentiate between debris cover and clean ice surfaces. It contains the
same accuracies and is homogeneous throughout the different Landsat scenes (Figure 13). The total
amount of debris-covered glacier area is 3580 ±798 km², i.e. 10% of the total glacierised area with small
differences among the four HMA regions. The lowest and highest area shares are found in Western (8%)
and Eastern Pamir (12%), respectively. There is no significant relation between glacier size and debris-
covered area share. The distribution in aspect is somewhat skewed towards North and North-East (12
and 11% vs. 8-9% in E, SE, S, SW, W, NW), but this is less of a systematic pattern than for the total
glacierised area. The highest values are found in Eastern Pamir where north-facing glaciers are debris-
covered by over 17%, whereas Pamir Alai exhibits the largest range (N = 15% vs. SW = 6%).


*Figure 13 : Debris cover on glaciers in the central Karakoram.*

Generally, there is no relation between the mean slope of a glacier and the area share of its debris cover.
However, the mean slope of the debris-covered part of the glaciers is 16.6° (±5.5) whereas the mean
slope of these glaciers is 26.1° (±3.2). This was expected since the debris cover is usually situated at the
flatter glacier tongues (Paul et al., 2004). Looking at the ablation area of all glaciers, the mean slope is
25.0° (±4.2). The ablation areas of more strongly debris-covered glaciers are somewhat flatter: glaciers
with a debris-covered area of ≥10% are on average 22.7° (±4.0) steep; in contrast, glaciers with less than
5% debris cover have a mean slope of 25.7° (±4.1).





## 6   Uncertainties and multiple digitising experiment

By applying previously found area uncertainties (±2.5% for clean ice, ±5% for debris-covered ice) to the mapped glacier area, the derived total glacier area is 35'287 ±1944 km². With the buffer method (clean ice $\pm^1/_2$, debris-covered ice ±1 pixel) we obtain a very similar uncertainty of ±1948 km². Both methods are applied to glacier complexes to avoid double counting of overlapping areas of adjacent glaciers. Finally, the multiple digitization experiment resulted in a ±13% standard deviation (averaged over all experiments). This value might seem rather high, but it reflects the mapping reality in challenging situations with debris-covered glacier tongues. For two of the three test regions, the difference between the largest and the smallest area mapped was less than 5% of the mean glacier area. The third example constitutes the case of a small (~2.9 km²) and steep glacier with a high share of its area hidden in shadow, a large and barely visible debris-covered part and adjacent rock glaciers (see Figure supplements). Here, the respective uncertainty is ±33%. Taking this as a worst-case scenario, only few such cases exist in a larger inventory and the high uncertainty has little impact on the overall uncertainty.

Paul et al. (2013 and 2015) showed that analyst interpretations for debris-covered glaciers and glacier parts in shadow can differ by up to 50%. Our experiment showed that if the glacier is affected by both shadow and debris cover and is additionally small, the differences can be even higher with up to 70%. The experiment also confirmed that area differences mainly depend on the interpretation of the debris-covered parts. Thereby, using coherence images improved the analyst's interpretation. Although the overall effect was small (on average ~1%), it reduced the dispersion of the analyst's interpretations considerably (see Figure 14). The different timing of Landsat (2000) and ALOS-1 PALSAR-1 (2007-2009) imagery had only a small impact, as geometric changes during these 7 years were small. The use of Google Earth imagery did not lead to notable outline modifications as they either had low quality (resolution, snow cover) or provided a mere confirmation of the existing interpretation from Landsat and coherence images. We conclude that the area uncertainty of the debris-covered parts of a glacier is in the order of 10 to 20%. However, at least one third of this uncertainty can be disregarded due to direct contact to clean-ice glacier parts (see Figure 13).

*Figure 14 : Results of the expert round robin, example glacier 2. (a) Shows mapping results solely based on the satellite image, whereas (b) shows mapping results after manual corrections using the additional source of coherence images and Google Earth hig- resolution imagery.*

The mapping uncertainty for the clean-ice glacier parts was found to be low, notwithstanding the simple method applied (constant threshold for all scenes). Using different thresholds of 2.0 ±0.3 yielded results



in the range of 5% of the debris-covered area, which is smaller than the uncertainty from the manual
correction of the debris-covered glacier parts.
All uncertainty values have to be seen in perspective to methodological uncertainties, e.g. the inclusion
of possible snowfields at high elevations, which can easily increase the area of a small glacier by 50% or
more. With this in mind, the uncertainties presented above are in general much smaller and are more of
an academic nature. As the uncertainties from the expert round robin are close to those from the buffer
method, we use the uncertainty derived by the buffer method as the uncertainties assigned to our
results, knowing that they are on the conservative side.
We also performed a comparison in regions where the CGI and the GAMDAM inventories are available, to
determine major differences among them. Compared to the CGI, our total glacier area is ~15% larger
(despite a similar glacier definition) and the CGI overlaps with 82% of our inventory. Our debris-covered
areas are somewhat larger along the margins of the tongues and more of the smaller glaciers at higher
elevations are included (Figure 15). Regions where the CGI area is larger (7% in total), are related to the
inclusion of areas enclosed by different branches of the same glacier, as well as dead ice and rock glaciers
in front of a terminus. The GAMDAM inventory covers 13% less area than ours and also overlaps with
82% of the area. Here the difference is clearly linked to a diverging glacier mapping definition, that mostly
excludes headwalls steeper than 40° (Nuimura et al., 2015). Moreover, many debris-covered glacier
areas and in some cases entire glaciers have not been mapped. On the other hand, almost all of the areas
covered by GAMDAM but not by our inventory are mapped as debris-covered glaciers. We think that
excluding steep headwalls leads to an incomplete inventory and that the inclusion of rock outcrops in
the CGI constitutes a commission error that need to be corrected for some applications. Overall, the
differing interpretation of debris-covered glacier parts is seemingly still the largest challenge and a main
source of differences in glacier extents for the same region when mapped by different analysts.

*Figure 15 : Comparison example of the three inventories.*

## 7    Discussion

Glacier mapping has come a long way in the last two decades with the positive result, that a globally
complete glacier inventory (RGI) of mostly high quality has been created (Pfeffer et al., 2014) and further
improvements are on-going (RGI Consortium, 2017). However, the issues presented above reveal that
achieving a high quality inventory requires a certain amount of manual effort and a solid glaciological
background, even if it is mostly used for correcting raw glacier outlines. Although several approaches



exist to automatically delineate debris-covered glaciers, their shortcomings (data availability,
time/effort, inhomogeneity of parameters) still outweigh the benefits and pure manual delineation was
found to provide the best results (Nagai et al., 2013).
The mapping of debris-covered ice was performed automatically by applying a single threshold value to
all scenes and subtracting the resulting clean-ice maps from the corrected outlines. This is likely the
easiest method that still provided very good results. Adapting the clean-ice threshold changed the
resulting debris-covered area only by ±2.5%, indicating that the transition from clean ice to complete
debris cover is relatively sharp. Due to the limited terrain shadow, we also had only a minor impact of
the changed threshold on area changes in shadow. This would be more critical in higher latitudes (e.g.
Paul et al., 2015). Herreid et al. (2015) used a function applied to Landsat band combinations that was
fit to manually derived reference data of a single glacier and adapted it to the various mapping dates
(using different Landsat sensors). This method might be superior, but it is more labour-intensive and
visual comparison with the figures in Herreid et al. (2015) show very high agreement. Another promising
approach applied by Kraaijenbrink et al. (2017) uses the normalised difference snow index together with
a composite image of Landsat 8 band 10 (thermal infrared) scenes, though it is sensible to detecting cast
shadows as debris cover and comes with the disadvantage of the much coarser (100 m) spatial
resolution. A major prerequisite for all these methods is the use of glacier outlines that are well adjusted
for debris cover. Glacier retreat was found to correlate with an increase in supraglacial debris cover (e.g.
Stokes et al., 2007) and hence, multi-temporal mapping of debris extent should be applied. Maybe the
single threshold value method applied here works as well. However, this implies limited overall
geometric changes of the related debris-covered glaciers to avoid a complete re-mapping. The debris
map itself reveals peculiarities such as large rock falls and glaciers that are strongly avalanche-fed and
can thus be a starting point for in-depth analysis of such phenomena. As extensive debris cover affects
glacier melt and geometry (e.g. Anderson and Anderson, 2016), we recommend including it in the
published glacier inventories (GLIMS, RGI), by (a) adding the debris mask as a polygon and (b) including
debris cover share in the attribute table.
When investigating glaciers in High Mountain Asia, the large area it covers gives the significance of debris
cover on glaciers. Up to now, there have been no reliable numbers of debris coverage for the entire
Pamir and Karakoram distinguished by a single method. Our results show a total of ~10% debris-covered
area, with many of the larger glaciers reaching 20% or more. These numbers complement and confirm
existing estimates in HMA that are based on smaller samples. Numbers reported from the central
Karakoram are 20% (Minora et al., 2016) and ~21% (Herreid et al., 2015), for the western Himalaya Frey
et al. (2012) give 16%, and for the entire Himalaya a ~10% coverage was calculated (Kraaijenbrink et al.,



2017; Bolch et al., 2012). The separately delineated debris-covered area provides a good basis for
investigations of debris cover changes or mass balance modelling.
The quality and availability of input data to compile an inventory has strongly increased over the past
few years, resulting in a higher credibility of the resulting outlines and topographic parameters as well
as a lower uncertainty of the glacierised area. When compiling a large-scale glacier inventory, it is
essential to have homogeneous quality of the input data used, at best also in a temporal sense. This can
be achieved by using globally consistent datasets such as the Landsat images and the GDEM2. However,
the latter does not fulfil the criteria of temporal consistency and future work might overcome this issue.
However, input data are never perfect. There are strategies like DEM fusion to improve DEM quality in
regions of very steep terrain or low contrast glacier surfaces (e.g. Shean et al., 2016; Lee et al., 2015;
Tran et al., 2014 and references therein), but the impact of such quality issues are difficult to assess
without accurately geo-referenced high-resolution reference data (Kääb et al., 2016; Frey and Paul,
2012). With the transparent automated processing line applied here and the few experts involved in the
manual corrections, we assume a homogeneous quality of the glacier outlines throughout the study
area. In any case, our glacier extents are on the one hand rather conservative in the debris-covered
ablation area leading to an underestimation of glacier area, and on the other hand include possible
perennial ice and snowfields in steep terrain at high elevations.
The pattern of glacier median elevations found in our study reflect combinations of climatic and
topographic aspects. A similar West-East and North-South gradient was also found in the study by Sakai
et al. (2015) who determined median elevations from a glacier inventory (GAMDAM, Nuimura et al.,
2015) for all of HMA. On the one hand, the latitudinal span of 7° decrease air temperatures and thus
median elevations towards the North; on the other hand, the precipitation decrease from West to East
due to lee-ward rain shadow effects increases median elevations in the Eastern Pamir and Karakoram.
Glacier median elevation is also linked to the topographic setting: in high-relief areas glaciers can extend
over a larger elevation range resulting in higher median elevations. Approximating the balanced-budget
ELA ($ELA_0$) with the median elevation has been successfully applied in many mountain ranges and works
well for different glacier types (Braithwaite and Raper, 2009). However, this concept does likely not apply
to surge-type glaciers as well as glaciers that are largely nourished by avalanches (Hewitt, 2011). For the
latter as well as debris-covered glaciers, $ELA_0$ values are expected to be higher than the ones we
calculated due to the additional accumulation and reduced ablation, respectively. This is supported by
the fact that we find debris-covered areas also above the median elevation and Braithwaite and Raper
(2009) mention possible accumulation-area ratio values below 0.5 for e.g. Himalayan glaciers.
We also performed a detailed analysis of uncertainties and analysed the most important sources
contributing to uncertainty. It is, however, impossible to retrieve an error as this would require a



comparison with appropriate reference data. The uncertainties presented here are based on different
methods and are partly higher than reported previously. This is mainly because of the high debris
coverage and the large number of (very) small glaciers. Under challenging conditions, area differences
among the analysts were as high as uncertainties due to the possible wrong consideration of seasonal
snow. Due to this, the total area of our inventory will likely be larger than other inventories for this region
as these might have excluded the maybe just snow-covered steep regions at highest elevations. Once
scenes without seasonal snow in these regions become available, glacier extents should be corrected
accordingly.
## 8 Conclusion
We have described how a new glacier inventory for a substantial part of western High Mountain Asia
(Karakoram and Pamir) has been created and presented in detail the derived characteristics of the
glaciers in this region. Special emphasis was given to the description of mapping challenges for debris-
covered glaciers (and distinguishing them from rock glaciers), seasonal snow, and shadow, along with
the selected solutions. In the absence of appropriate reference datasets, we applied various methods
for uncertainty assessment and compared our outlines to other existing inventories covering the same
region. As an extension to already existing datasets we included outlines and percentages of the debris-
covered area for each glacier.
Overall, we mapped 27437 glaciers covering 35287 ±1209 km² of which ~10% were debris covered. The
ASTER GDEM2 was found superior over the SRTM DEM (1 arc second) to derive drainage divides and
topographic information for each glacier as the later suffered from too many (wrongly interpolated) data
voids in this region. The application of a constant band ratio threshold to derive clean-ice areas for all
scenes to create the debris-cover maps was found to be very robust. Uncertainties derived from three
different methods were all in good agreement (3.4%) but the multiple-digitizing experiment also
revealed larger deviations among the analysts under challenging conditions (debris, shadow). Clearly,
the availability of coherence images improved the quality and consistency of the manual corrections for
debris-covered glaciers considerably.
The analysis of the topographic information revealed several interesting dependencies among the
glaciers and also across the regions. Despite the fact that in the Karakoram the largest glaciers are facing
SE (Siachen, Biafo), E (Batura, Skamri) or W (Baltoro, Hispar), most glacier area (47%) is still exposed to
the three northern sectors. Glacier median elevation has little dependence on aspect but a strong one
on longitude and latitude (higher towards the drier north and east), indicating a close relation to
precipitation amounts. Glacier hypsometry reveals a peak distribution that is highest (~5700 m) in the



Karakoram, similar but 700 m lower in Eastern and Western Pamir, and lowest in Pamir Alai (~4200 m).
Glaciers in the Karakoram have a comparably higher area share at lowest elevations and glaciers larger
5 km² or debris-covered glaciers are flatter (22.6° and 16.6°, respectively) than in average (26.4°). By
location, glaciers are especially flat (<15°) in their lowest third and progressively steeper (>30°) in the
uppermost third, indicating the dominance of large valley glaciers with very flat tongues and steep head
walls. Both, glacier outlines and the separate outlines of the debris-covered parts are freely available
from the GLIMS database.

**Acknowledgements**
We acknowledge the contribution of the ESA projects Glaciers-cci (4000109873/14/I-NB) and Dragon 4
(4000121469/17/I-NB).The manual digitizing experiment was performed by the authors and additional
contributions by H. Frey and R. Le Bris.





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



**Figures**

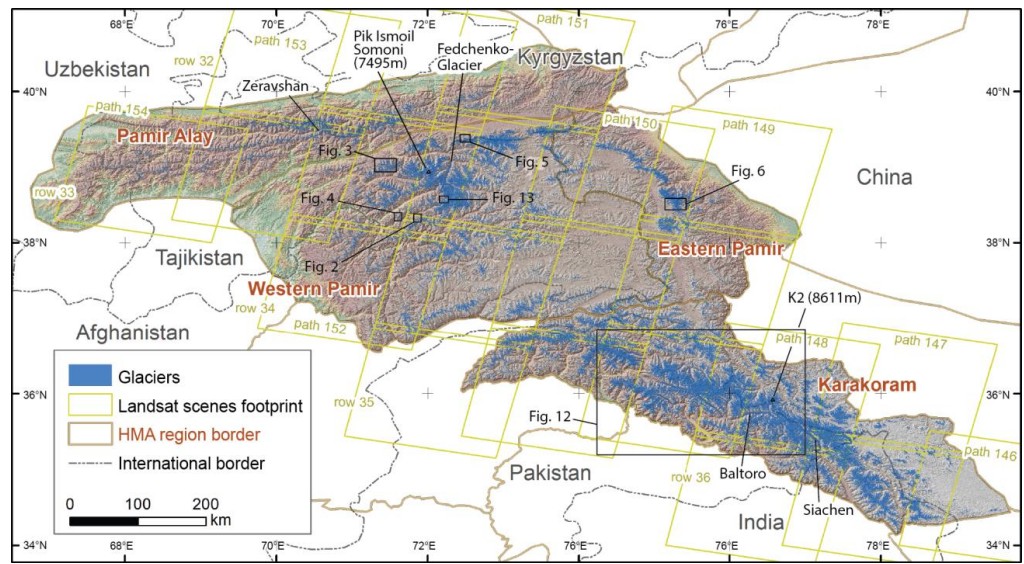

Figure 1 : The study region in HMA covers four mountain ranges. Annotations denote locations of figures in the paper and points of orientation. International borders are tentative only as they are disputed in several regions.

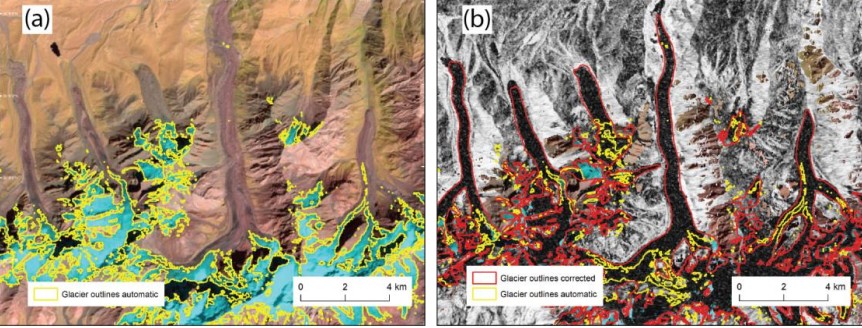

Figure 2: Mapping of heavily debris-covered tongues using PALSAR-1 coherence images.




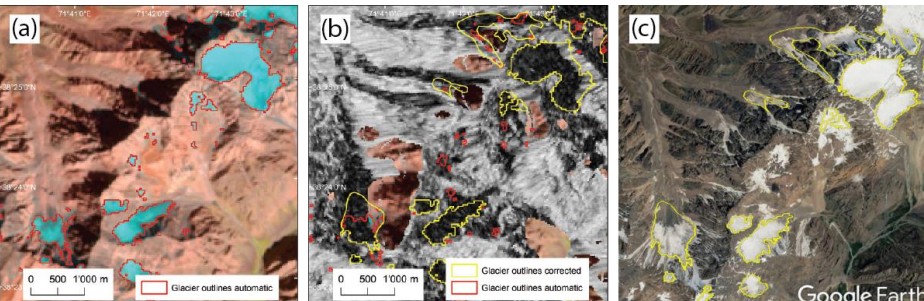


Figure 3: Elongated rock glaciers that are (almost) connected to the active glacier tongue are hard to distinguish. PALSAR-1
coherence images are not decisive in this case, but high resolution imagery is.

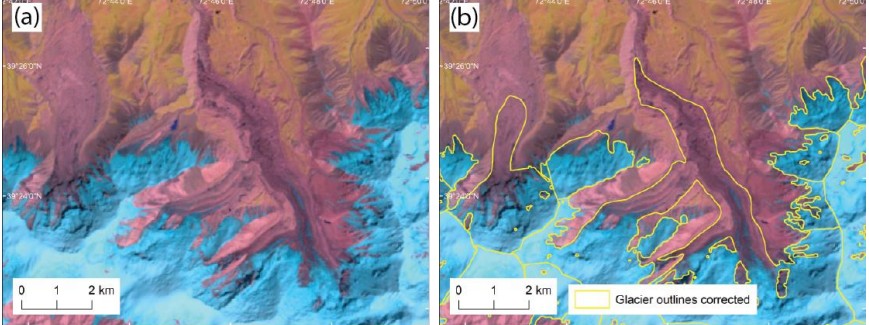


Figure 4: Extensive moraines and large areas of debris can be found on dead ice and active glaciers.

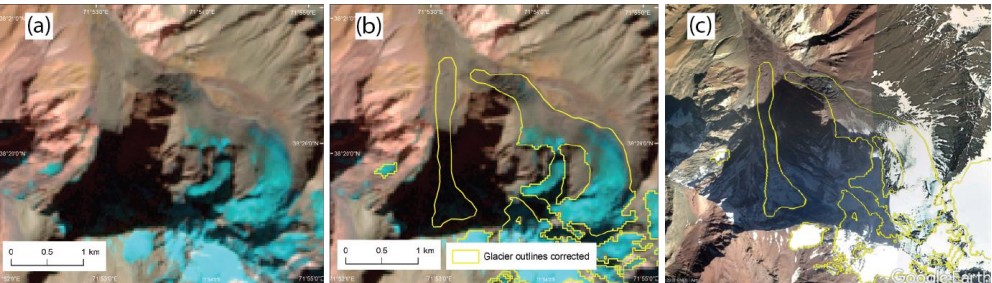


Figure 5: Glacier detection in shadow with the supporting input of high resolution Google Earth images.



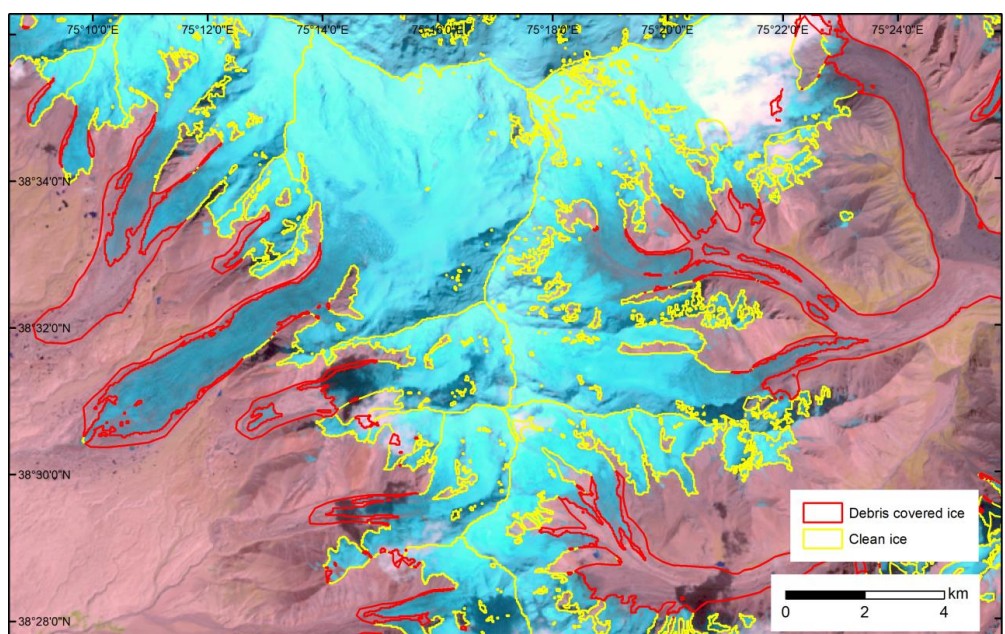


Figure 6: Debris cover classification in the Kongur Shan in the Eastern Pamir.


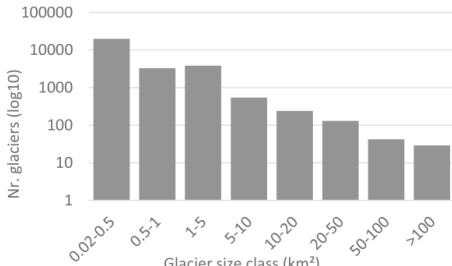


Figure 7: Histogram of all glaciers by number. Please note the logarithmic scale of the y-axis.


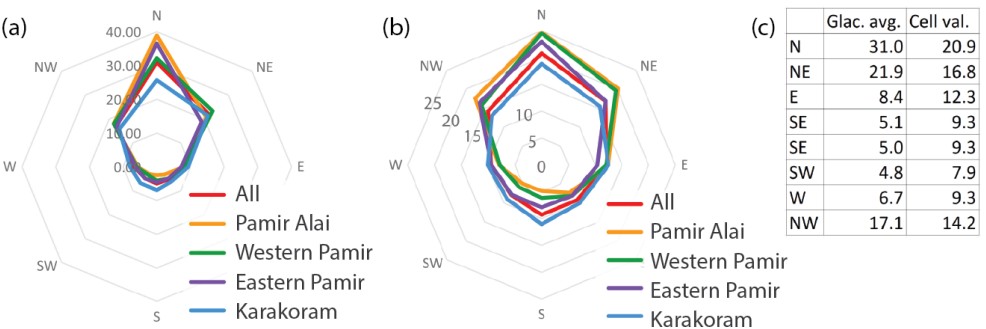

|  | Glac. avg. | Cell val. |
|---|---|---|
| N | 31.0 | 20.9 |
| NE | 21.9 | 16.8 |
| E | 8.4 | 12.3 |
| SE | 5.1 | 9.3 |
| SE | 5.0 | 9.3 |
| SW | 4.8 | 7.9 |
| W | 6.7 | 9.3 |
| NW | 17.1 | 14.2 |






*Figure 8: Glacier orientation of the different HMA regions. (a) shows the values based on average glacier aspect, (b) is based on*
*the 30 m raster cells. Lower elevations tend to have a higher share of north-facing glacier area. The respective numbers of "All"*
*are given in the table (c).*

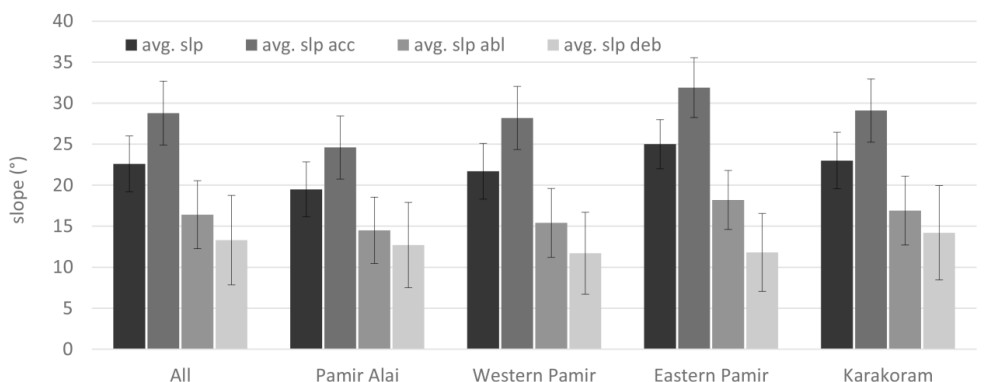


*Figure 9: Slope per glacier regions and surface type (avg. slp = average slope of glacierised area, avg. slp acc = average slope in*
*the accumulation area, avg. slp abl = average slope in the ablation area, avg. slp deb = average slope in the debris-covered area).*

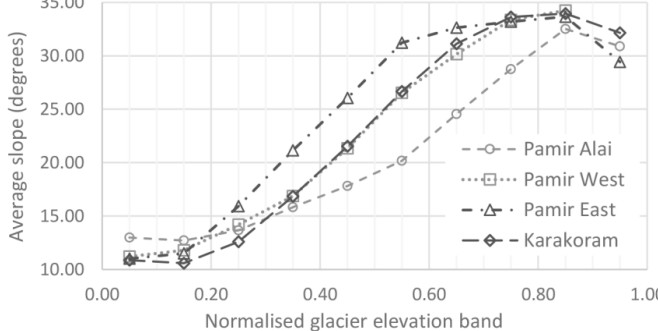


*Figure 10: Glacier slope along ten glacier elevation sections. The glaciers were normalised for elevation to compare high and low*
*elevation glaciers.*





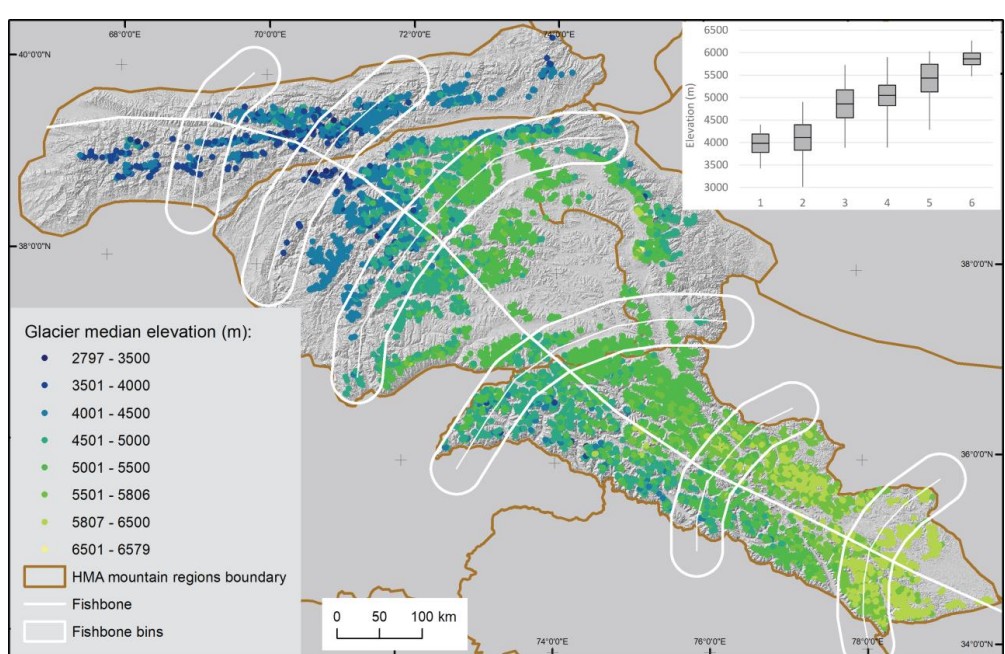


*Figure 11: Glacier median elevation over the study area of glaciers larger than 0.5 km². The inset shows median elevation,*
*standard deviations and minimum and maximum elevations per bin.*



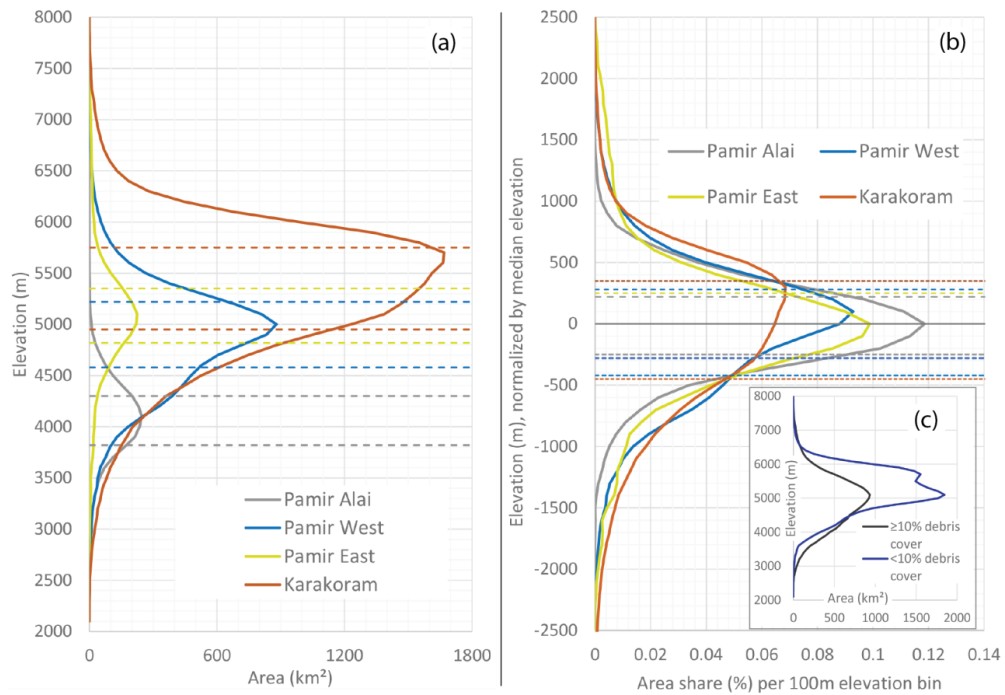


Figure 12: Glacier hypsography of the different regions (a), normalized by the respective median elevation (b). Dashed lines
represent the 25% and 75% area elevations. (c): Hypsography comparison of more and less debris-covered glaciers.





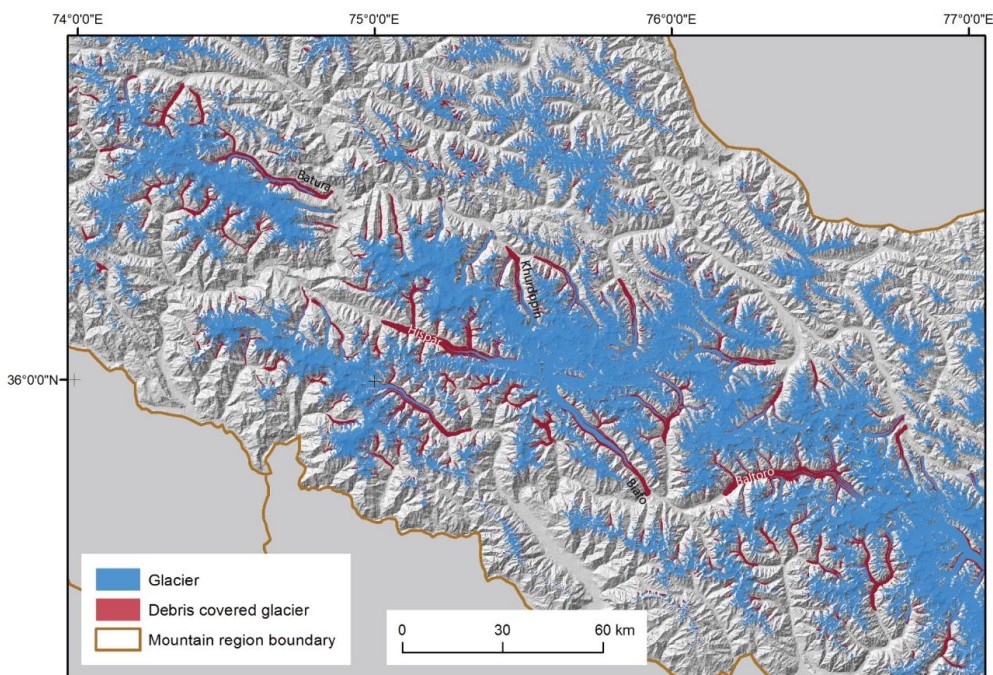


*Figure 13: Debris cover on glaciers in the central Karakoram.*

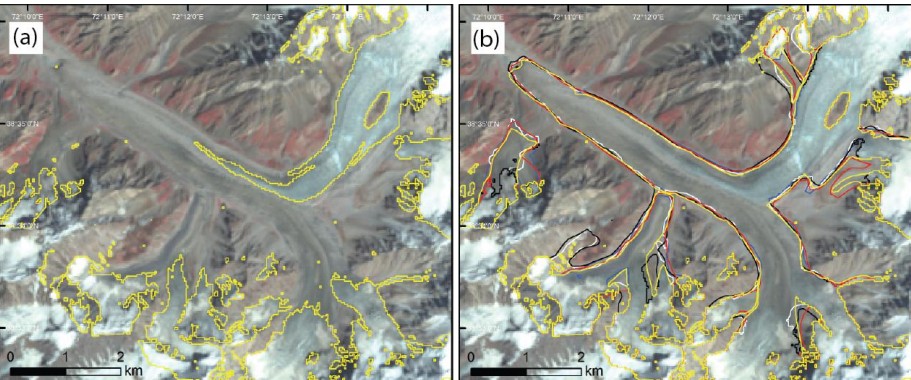


*Figure 14: Results of the expert round robin, example glacier 2. (a) Shows mapping results solely based on the satellite image,*
*whereas (b) shows mapping results after manual corrections using the additional source of coherence images and Google Earth*
*hig- resolution imagery.*



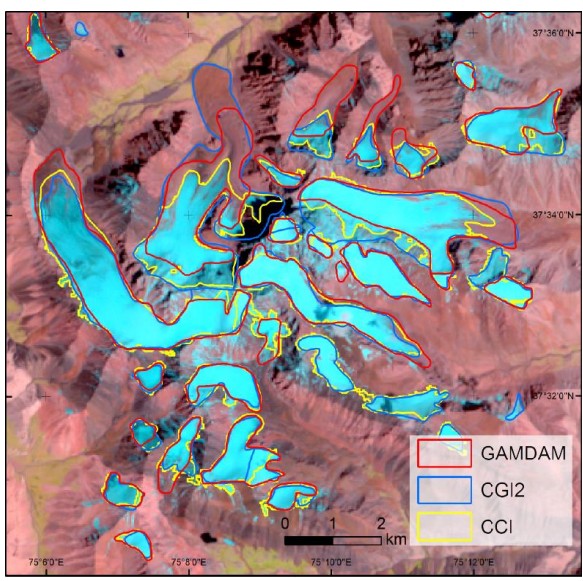


*Figure 15: Comparison example of the three inventories.*




**Dataset**
Dataset is downloadable at
http://www.geo.uzh.ch/~nmoelg/glacier_inventory.zip
