# Peer review of "DERIVED FROM LANDSAT DATA: DISTRIBUTION OF DEBRIS COVER AND MAPPING"

_Earth System Science Data, 2018_

## Referee Comment (RC1) · Anonymous Referee #1 · 3 May 2018

General Notes The paper aims at generating a consistent glacier inventory for the glacierised mountain ranges of Pamir Alai, Western and Eastern Pamir and Karakoram for the year around 2000, using Landsat TM/ETM+ images of around 2000 and ALOS-1 PALSAR-1 data of around 2007. Additionally, the manuscript also attempts to highlight the mapping challenges encountered in the study region. The manuscript is well-written, methods are well-presented and uncertainties are defined in detail. The outcomes of the work are of significant scientific interest. However, few points mentioned in specific notes to authors need due consideration.

Specific notes to authors Page 4 Line 79-83: 'Key challenges. . ...study region'. These

issues have not been tackled any differently in the manuscript. Page 8 Line 186: 'GDEM2' should be replaced by ASTER GDEM2 throughout the manuscript. Page 10 Section 4.2: Much of the details and information given in this section are already known to the glaciology community and authors do not even offer any new solution to these issues. For instance, author indicate that usability of the ALOS-1 PALSAR-1 coherence images decreases with the decreasing glacier sizes (Page 10 Line 246-247). Nevertheless, their results indicate that about 83% glaciers in the study region are < 1km2. This leaves authors with the only solution i.e. very high-resolution google earth imagery. Again, in the case of rock glaciers, google earth images were used as an active source (Page 11 Line 280-281). So Section 4.2 can be either obliterated or merged with Section 4.1. Page 11 Line 268-284: Since rock glaciers were separated from debris-covered glaciers, their status may be quantified. Page 11 Line 287-289: 'For larger glaciers….consistent glacier outlines'. What is meant by this? When the debris-covered portion is mapped/corrected by using 2007-09 coherence data then what about temporal consistency. A proper discussion needs to be added. Page 12 Line 296-297: 'Despite…seasonal snow'. By how much %? At least a rough estimate may be provided. Page 12 Line 298-310: Why a topographic correction method has not been applied to minimize the shadow effect? Page 12: Section 4.3: If the clean ice/snow area were mapped using band ratio and debris covered parts were delineated manually, one can expect that the debris cover area is readily measureable. Then why a separate methodology has been adopted to calculate debris-covered area share of the glaciers? Page 13 Line 329: Replace 'Chapter 6' with 'Section 6'. Page 13 Line 343-344: 'Snow fields…..as glaciers'. Why? Inclusion of seasonal snow patches instead of perennial snow introduces large errors. Therefore, multitemporal analysis is recommended to separate seasonal snow from perennial snow or glaciers (Paul et al., 2009). Page 13 Line 354-355: 'We assigned…regenerated glaciers'. Meaning not clear. Page 14 Line 384-387: What about the PALSAR images of 2007 to 2009. Here authors talk only about temporal consistency of Landsat images but do not consider/quantify the temporal uncertainties stemmed from correction of debris-covered

part using coherence data which have a considerable temporal separation. Page 20 Line 556-557: 'The different....were small'. This seems to be an over simplified statement. The changes in glacier geometry over the period i.e. 2000$\pm$2 to 2007$\pm$2 need due consideration and discussion. Page 21-24 Line 597-676: The discussion section is quite weak and should be strengthened. Only the last paragraph (line 652-666) discussed some interesting ideas.

---

## Referee Comment (RC2) · J. G. Cogley (Referee) · 15 May 2018

**Comments on Mölg, N. et al., A consistent glacier inventory for the Karakoram and Pamir region derived from Landsat data: distribution of debris cover and mapping challenges, submitted to *Earth System Science Data Discussions***
*Graham Cogley, May 2018*

*General Comments*
This paper describes in detail the compilation of an inventory of all the glaciers in the Karakoram, Pamir and Pamir Alai in western High Mountain Asia. The authors emphasize (rightly) the consistency of their methods across the region, and note that the results are intended for incorporation into two global inventories, GLIMS and the Randolph Glacier Inventory. Careful treatment is given to the exclusion of cloud cover, the identification of debris-covered ice (assisted by the use of radar-interferometric imagery), and the digitization of drainage divides between glaciers. The authors note that they are unable to estimate "accuracies" in the absence of reference data representing "truth", but I do not see this as a serious shortcoming. For one thing it is unavoidable, and for another I think they have done a very good job of estimating "uncertainties". The only point on which their error analysis becomes unsatisfactory is the estimation of seasonal snow cover wrongly included within their glacier outlines, but this is a problem that is currently intractable and their attempts to exclude seasonal snow strike me as being equal to the best that I have seen in other studies.

This is an excellent paper, and could serve as a model for other glacier inventories, especially those which involve large numbers of glaciers. Having said that, the manuscript is not entirely free of ambiguity and some contradiction, but I would recommend acceptance if the authors consider carefully my substantive comments below. My stylistic comments are minor but fairly numerous; I think they will improve readability and therefore impact on readers.

*Substantive Comments*

L43      "and related inventories" sounds odd. Perhaps expand to something like ", and their accompanying attributes as recorded in glacier inventories,". But the real problem seems to be that this sentence sounds as though it is about outlines but is really about inventories.

L57      Perhaps you could find a way to add a mention that version 6.0 of the RGI is unchanged in HMA.

L121-145      Considering that this is a data paper, I suggest that this paragraph could be omitted as going beyond the scope of the manuscript. The only point that might be worth making in the paragraph above is that glacier climate is very poorly known because most weather stations are at lower elevation.

L248      A remark should perhaps be added here to note that the coherence images do not allow the identification of stagnant ice. And do you regard stagnant ice as part of the glacier, or not? Apparently yes, to judge by L337-338. Of course little is known about the extent of glacier ice that has stopped moving, but it is by no means unknown.

L298      Change "partly" to "sometimes". But is this correct? The problem is not that the reflectivity of the surface is near zero, but that the incident irradiance is near zero. I think it would be correct to say that the reflected *radiance* is near zero.

L351,L603      It sounds as though the automatic algorithms for drainage divides reduced the processing time by about a tenth. As one who has digitized rather a lot of glacier outlines manually, I agree that it is a time-consuming exercise, but I cannot but wonder whether the automatic aids were worth the trouble. At L603 you say they were, but then immediately contradict yourselves by saying that manual work gave the best results.

L362      Surely the uncertainty also depends on pixel size (with the number of pixels becoming small as you look at smaller glaciers)?

L386-387      At least when calculating rates of change, you can reduce the temporal uncertainty to zero if you use exact dates for calculating $\Delta t$, as explained by Cogley (2016, *Annals of Glaciology*, **57**(71),

41-49). I agree that if you do this then differences between $t_0$s or $t_1$s will not matter if they range over only a couple of years.

L573     I know there is no satisfactory answer to the question of uncertainty due to seasonal snow, but can you suggest even a semi-quantitative estimate (like this one for a hypothetical small glacier) for the entire inventory?

L580     Unless you clipped your inventory against the frontier of China, I do not think this comparison is of much value.

L591     "still the largest challenge": only because all the inventories have the same kind of trouble with seasonal snow.

L625-627     I agree completely with the recommendation to include debris cover in inventories, but it will be a long time before the data approach global completeness.

L686     Please clarify. This says that about 2744 glaciers had debris cover, but I suspect you mean that about 10% of the glacierised area is debris-covered (or is it 5%, as in the Abstract?).

*Stylistic Comments*

L15     Delete initial "The".

L22     Change "way" to "method".

L22-25     Break up this long sentence with semicolons after "information" (L24) and "GDEM2" (L25).

L29     For some reason the comma separators in all numbers in the manuscript have come out as apostrophes.

L31     I would begin a new sentence at "this". But, more importantly, what is "this"? The range of median elevations? The range of all elevations? Clarify, e.g. by saying "The <whatever> is largely due …".

L46     No space before "Sakai".

L47     Delete "also".

L50-52     "When glacier outlines are of poor quality, related … have higher uncertainties.". "Has improved significantly in recent years, during which … have been published,".

L54     "the second Chinese".

L55     "only partially considered ice cover on steep".

L60     Change "comprise" to "suffer from".

L68     I find "partly heavily" rather strange. "heavily" is probably not necessary.

L72     "supraglacial" (no hyphen). "from that of clean ice".

L75-76     "higher and more extensively glacierised". (I am not sure what a "large" mountain range is.)

L81     Another instance in which semicolons are needed: after "main glacier" and "glacier ice".

L96     "with K2".

L97-98     Garbled sentence? I think it means "The central Karakoram and inner Pamir are two … worldwide, and include some extremely large glaciers …".

L100     Delete the repetitive sentence "The study region …".

L102-103     Move "also" to before "present".

L146     "Glacier".

L148     "that was first identified by Hewitt".

L151     "1970s" (no apostrophe).

L159     Change "like" to "as".

L163     Change "are" to "have been".

L179     Comma after "interference".

L193     "to work only with". "many fewer data voids".

L207     "which is smoothed".

L254-257     Sentence not very clear. I think "the importance of" should be deleted, and "are considered" should be changed to "need to be considered". But please clarify.

L283     "In the case that".

L295     "(with less snow but possibly more cloud)".

| | |
|---|---|
| L304 | "contrast-enhanced". |
| L307 | Delete "in this region". |
| L315 | "were available for most of the study region". |
| L329 | "by less". "section 6". |
| L343 | "be they". Comma after "perennial". |
| L346 | "two steps". |
| L350-351 | "in general glaciers are divided by very steep …". |
| L365 | Delete "G." (His initials are A.G. in any case.) |
| L381-382 | "substantial debris-covered parts; they also feature difficulties …". Change "confluences regions" to "confluences". |
| L390 | Semicolon, not comma. "larger than". Same at L399, L403, L403 (check throughout, in fact; there are very many instances). |
| L400 | "glacierised". |
| L404 | "by only a few metres". |
| L413 | I think "to the south" should be "from the south". |
| L425 | Change "small-sized" to "small". |
| L465 | Delete "by". |
| L485-486 | Italicize $p$ and change "R" to $r$ (lower-case italic) $r$. $R$ is usually reserved for correlations from multivariate regression. |
| L518 | Change "accuracies" to "uncertainties". (You said earlier, at L358, that you do not calculate accuracies.) |
| L538 | "previously assumed". |
| L544 | Should "regions" be "glaciers"? |
| L546 | Change "constitutes the case of" to "is". |
| L566 | "high-". |
| L572 | "from the perspective of". |
| L590 | "needs". |
| L597 | Move the comma to follow "decades". |
| L614 | "shows". |
| L616 | "sensitive". |
| L617 | "of  a much coarser". |
| L628 | I do not understand the "large area …" part of this sentence. Please clarify, or perhaps just delete the sentence, which sounds rather bland. |
| L642 | "criterion" ("criteria" is plural.) |
| L645 | "is difficult". |
| L652 | "reflects". |
| L655 | "decreases". |
| L661 | "likely does not". |
| L668 | "a rigorous error". (I think you have done a pretty good job without reference data.) |
| L670 | "and some are higher". |
| L679 | ", and have presented in detail". |
| L687 | "superior to". |
| L688 | "the latter". |
| L703 | "on average". |
| L706 | Delete the comma. |
| L711 | "with additional contributions". |
| L714 | I have not checked the references exhaustively, but several need to have paper titles without initial capitals, and the abbreviation (or not) of journal titles needs to be checked. |
| L751 | Missing right parenthesis. |
| L832 | " and Fountain, A.G.:". |

---

## Author Comment (AC1) · 4 Jul 2018

Submission 1: A consistent glacier inventory for the Karakoram and Pamir region derived from Landsat data: Distribution of debris cover and mapping challenges.

Response to reviewer's comments.

NOTE: You can find the same text attached as pdf file.

Response reviewer #1

We thank reviewer #1 for the constructive and critical feedback. We have addressed all of the comments individually as listed below.

[Figure]

Comment #1, concerning Page 4, Lines 79-83:

'Key challenges...study region'. These issues have not been tackled any differently in the manuscript.'

We have described in section 4.4 (L340) that we separated Bivachny glacier from Fedchenko although it is a connected tributary. To be clear, we have now also added in this section that we left surge type tributaries connected to a larger main glacier for the sake of consistency with earlier datasets and the GLIMS definition and that a dataset where all these (short-term) tributaries are separated is in preparation.

Comment #2, concerning Page 8, Line 186:

'GDEM2' should be replaced by ASTER GDEM2 throughout the manuscript.'

We have replaced 'GDEM2' by 'ASTER GDEM2' when first mentioning the dataset and added a reference to 'GDEM2'.

"Additionally, we have also used coherence images derived from ALOS-1 PALSAR-1 scenes acquired around 2007 to aid in mapping the debris-covered glacier parts, and the global digital elevation model GDEM Version 2 from ASTER (hereafter referred to as GDEM2). "

Comment #3, concerning Page 10, Section 4.2:

'Much of the details and information given in this section are already known to the glaciology community and authors do not even offer any new solution to these issues. For instance, author indicate that usability of the ALOS-1 PALSAR-1 coherence images decreases with the decreasing glacier sizes (Page 10 Line 246-247). Nevertheless, their results indicate that about 83% glaciers in the study region are < 1km2. This leaves authors with the only solution i.e. very high-resolution google earth imagery. Again, in the case of rock glaciers, google earth images were used as an active source (Page 11 Line 280-281). So Section 4.2 can be either obliterated or merged with Section 4.1.'

We agree that there is not much news in the methodological approach and we apply here well-established techniques. We think this is justifiable in the context of this journal that has a focus on the accurate description of the methods applied to create a dataset rather than introducing new scientific approaches. It is also true that most glaciers are small and often coherence images cannot be applied to correct the outlines of these small glaciers. However, for the many medium- to large-sized glaciers this is a necessity and in several regions (e.g. northern Pamir) this information is almost indispensable. Coherence images are still the preferred method for their detection and we propose this method as the main solution to overcome the described mapping difficulties. The following section about the challenges of coherence images was shortened without reducing the information content.

Comment #4, concerning Page 11, Lines 280-281:

See comment #3. The problem of separating rock glaciers from debris-covered glacier tongues is a difficult one. In our opinion, it is often not sufficiently accounted for in the methods sections of other studies. We think that the fact that (active) rock glaciers are separate cryogenic features that are different from glaciers and non-cryogenic land surface features justifies their discussion in the manuscript. As described in the text, they can be recognised by certain surface characteristics and the use of coherence and high-resolution images. Therefore, we would prefer keeping this section about the rock glacier challenges and the respective mapping solution in the text.

Comment #5, concerning Page 11, Lines 268-281:

Since rock glaciers were separated from debris-covered glaciers, their status may be quantified.

We fully agree that an inventory of rock glaciers would be a very valuable dataset. However, the goal of our study was to give on overview specifically of the glaciers of the region and a proper analysis of their characteristics. Additionally to the separation of glaciers and rock glaciers that are in direct contact, all individual rock glaciers in

glacier-free basins would need to be mapped, and a separate study is needed to locate and delineate them in a consistent and complete way.

Comment #6, concerning Page 11, Lines 287-289:

'For larger glaciers…consistent glacier outlines'. What is meant by this? When the debris-covered portion is mapped/corrected by using 2007-09 coherence data then what about temporal consistency. A proper discussion needs to be added.

We agree that the sentence is difficult to understand and will rewrite it. To be clearer, the coherence images are primarily used for a general identification of debris-covered glacier tongues (and tributary connections) rather than the complete delineation. The exact positioning – in particular in the terminus region – is done based on the original image but guided by the coherence image. This will be better described in the text.

Comment #7, concerning Page 12, 296-297:

'Despite…seasonal snow'. By how much %? At least a rough estimate may be provided.'

Considering a time period of +/- ten years around the year 2000, the scenes we have chosen are among the ones with least snow cover. We fully agree that an estimate of the uncertainty introduced by also mapping seasonal snow as glaciers would be a valuable contribution to the uncertainty assessment. We have discussed several possibilities but have come to the conclusion that none of these can yield reliable numbers on which to base an estimate; we therefore refrained from such an attempt. Nevertheless, it would be a very useful exercise and we are open for ideas and suggestions!

Comment #8, concerning Page 12, Lines 298-310:

'Why a topographic correction method has not been applied to minimize the shadow effect?' With the band ratio including the additional band 1 threshold snow and ice can also be accurately detected and mapped in shadow regions.

Comment #9, concerning Page 12, Section 4.3:

If the clean ice/snow area were mapped using band ratio and debris covered parts were delineated manually, one can expect that the debris cover area is readily measureable. Then why a separate methodology has been adopted to calculate debris-covered area share of the glaciers?'

This question is fully justified and can at best be answered by the fact that not only debris cover has been corrected manually but also some regions in shadow or clouds. These regions can be largely excluded by the separate methodology that restricted the debris calculation to the ablation area of the glaciers (below their median elevation), as debris is primarily found here.

Comment #10, concerning Page 13, Line 329:

Replace 'Chapter 6' with 'Section 6'. Done.

Comment #11, concerning Page 13, Lines 343-344:

'Snow fields...as glaciers'. Why? Inclusion of seasonal snow patches instead of perennial snow introduces large errors. Therefore, multitemporal analysis is recommended to separate seasonal snow from perennial snow or glaciers (Paul et al., 2009).'

Yes, we fully agree (at least when images with better snow conditions exist). As explained on Page 11/12, Lines 285-297, we have thus applied both a size filter as well as multi-temporal images in order to limit the inclusion of seasonal snow to the lowest amount possible. In Lines 343-344 we just want to say that remaining automatically classified polygons are considered as glaciers. The formulation was probably unclear, so we slightly changed the sentence and added an explaining sentence:

"Automatically classified polygons larger than this are considered as glaciers in this inventory, but this does neither mean that all seasonal or perennial snow fields have been excluded, nor that some of the mapped glaciers are in fact perennial snow."

Comment #12, concerning Page 13, Lines 354-355:

'We assigned. . .regenerated glaciers'. Meaning not clear.'

We reformulated the sentence and hope it is clear now:

"We manually assigned the same ID to separated glacier polygons that were obviously linked by mass transport (e.g. regenerated glaciers)."

Comment #13, concerning Page 14, Lines 384-387:

'What about the PALSAR images of 2007 to 2009. Here authors talk only about temporal consistency of Landsat images but do not consider/quantify the temporal uncertainties stemmed from correction of debris-covered part using coherence data which have a considerable temporal separation.'

As mentioned above, we fully agree with you about the much larger temporal difference for the PALSAR scenes. As explained in comment #6, we will add a paragraph clarifying the issue.

Comment #14, concerning Page 20, Lines 556-557:

'The different...were small'. This seems to be an over simplified statement. The changes in glacier geometry over the period i.e. 2000_2 to 2007_2 need due consideration and discussion.'

Yes, agreed. Without further explanation this statement appears strange. We hope that the added explanation (see comments #6 and #13) will clarify why the 7 years difference have limited impact on the temporal consistency.

Comment #15, concerning Pages 21-24:

'The discussion section is quite weak and should be strengthened. Only the last paragraph (line 652-666) discussed some interesting ideas.'

We agree that there is potential for condensing the discussion and will rewrite the

related parts. Our focus was on a critical review of our methods and results in the light of existing studies and what the results might imply. This is maybe beyond of what is required for this journal but we think it is still useful additional information for the reader.

 

Response reviewer #2

We thank J. Graham Cogley for this thorough review and are happy about the positive impression. We have addressed all the substantive and stylistic comments as stated below.

A. Substantive comments:

Comment #1, Line 43:

"and related inventories" sounds odd. Perhaps expand to something like ", and their ac­companying attributes as recorded in glacier inventories,". But the real problem seems to be that this sentence sounds as though it is about outlines but is really about inven­tories.

Agreed, we will change the sentence to:

"Glacier outlines and their accompanying attributes as recorded in glacier inventories provide the baseline for climate change impact assessments. . ."

Comment #2, Line 57:

'Perhaps you could find a way to add a mention that version 6.0 of the RGI is unchanged in HMA.'

Yes of course, the sentence will be changed to:

"As both inventories have been combined for version 5.0 and transferred unchanged to version 6.0 of the Randolph Glacier Inventory (RGI). . ."

Comment #3, Lines 121-145:

'Considering that this is a data paper, I suggest that this paragraph could be omitted as going beyond the scope of the manuscript. The only point that might be worth making in the paragraph above is that glacier climate is very poorly known because most weather stations are at lower elevation.'

We fully agree that this background is rather specific for a data paper and will condense it somewhat. We would like to keep some key information as we have later on some discussion about the spatial variability of median elevation and its relation to precipitation amounts. This is also a larger point in some recent studies (e.g. Sakai et al. 2015) and the 'Karakoram Anomaly' in general.

Comment #4: Line 248:

'A remark should perhaps be added here to note that the coherence images do not allow the identification of stagnant ice. And do you regard stagnant ice as part of the glacier, or not? Apparently yes, to judge by L337-338. Of course little is known about the extent of glacier ice that has stopped moving, but it is by no means unknown.'

Actually, the coherence images also allow identification of stagnant ice (to some degree) as coherence loss is not only due to movement but also due to other factors influencing the backscatter (e.g. melt at exposed ice cliffs, chaotic movement of debris). In general, we have considered stagnant ice masses as part of the glacier when they are physically connected. However, ice masses being remnants of a previous surge were excluded when a separate terminus could be identified (see Lines 337-339).

Comment #5, Line 298:

'Change "partly" to "sometimes". But is this correct? The problem is not that the reflectivity of the surface is near zero, but that the incident irradiance is near zero. I think it would be correct to say that the reflected radiance is near zero.'

We will change 'partly' to 'sometimes' as it largely depends on the atmospheric conditions (i.e. diffuse irradiance) that change with the day of image acquisition.

Comment #6, Lines 351, 603:

'It sounds as though the automatic algorithms for drainage divides reduced the processing time by about a tenth. As one who has digitized rather a lot of glacier outlines manually, I agree that it is a time-consuming exercise, but I cannot but wonder whether the automatic aids were worth the trouble. At L603 you say they were, but then immediately contradict yourselves by saying that manual work gave the best results.'

Here our statement was likely misleading. We only say that 10% of the total processing time is taken by the automated methods (e.g. selecting the correct thresholds) and 90% is required for manual editing. But the latter is only applied to maybe 5% of the total length of the perimeter. So a complete manual digitization would take about 20 times more time (roughly estimated, likely much more). For the drainage divides the algorithm is fully automated and really required to have consistency in the upper parts. In our experience merging the lower parts of the automatically created divides is likely as time-consuming as digitizing them manually, but again we here prefer the consistency of the dataset compared to a complete manual digitisation.

Comment #7, Line 362:

'Surely the uncertainty also depends on pixel size (with the number of pixels becoming small as you look at smaller glaciers)?'

Yes, the area uncertainty increases towards smaller glaciers and is not a constant value. As the value is smaller for larger glaciers, the 2-5% is likely a good average for a larger glacier sample (comprising the full range from small to large glaciers).

Comment #8, Lines 386-387:

'At least when calculating rates of change, you can reduce the temporal uncertainty to zero if you use exact dates for calculating $\Delta t$, as explained by Cogley (2016, Annals of Glaciology, 57(71), 41-49). I agree that if you do this then differences between t0s or t1s will not matter if they range over only a couple of years.'

[Figure]

We agree that for change assessment it would be preferable to use the exact dates as provided in the attribute table of the inventory. For other, more general applications such as regional-scale hydrological modelling one can safely go ahead with assigning a year 2000 to all glacier outlines of this dataset, but it certainly depends on the application.

Comment #9, Line 573:

I know there is no satisfactory answer to the question of uncertainty due to seasonal snow, but can you suggest even a semi-quantitative estimate (like this one for a hypothetical small glacier) for the entire inventory?

Considering a time period of +/- ten years around the year 2000, the scenes we have chosen are among the ones with least snow cover. We fully agree that an estimate of the seasonal snow would be a valuable contribution to the uncertainty assessment. We have thus discussed several possibilities of determination but have come to the conclusion that none of them can yield reliable numbers on which to base an estimate (e.g. the value will very sensitively depend on the size of the 'real' glacier). Hence, we refrained here from such an attempt. Nevertheless, it would be a useful exercise and we are open for ideas and suggestions!

Comment #10: Line 580:

'Unless you clipped your inventory against the frontier of China, I do not think this comparison is of much value.'

Fully correct, i.e. the comparison has indeed only been performed for the common region. The text will be adjusted to be clear.

Comment #11, Line 591:

"still the largest challenge": only because all the inventories have the same kind of trouble with seasonal snow.

Yes, agreed. We will write:

"Overall, the differing interpretation of debris-covered glacier parts and seasonal snow are seemingly the main source of differences in glacier extents for the same region when mapped by different analysts."

Comment #12, Lines 625-627:

'I agree completely with the recommendation to include debris cover in inventories, but it will be a long time before the data approach global completeness.'

As far as we can say it from discussions with colleagues, such a dataset will appear soon.

Comment #13, Line 686:

'Please clarify. This says that about 2744 glaciers had debris cover, but I suspect you mean that about 10% of the glacierised area is debris-covered (or is it 5%, as in the Abstract?).'

We wanted to say that ∼10% of the total area is debris-covered. The 5% number stated in the abstract is incorrect and will be changed. We'll also change the sentence in Line 686 to make it clear:

"Overall, we mapped 27437 glaciers covering 35287 ±1209 km$^2$, about 10% of the area being debris-covered."

B. Stylistic comments We thank the reviewer for the detailed stylistic comments and will address them as suggested.

Please also note the supplement to this comment:
https://www.earth-syst-sci-data-discuss.net/essd-2018-35/essd-2018-35-AC1-supplement.pdf